# EFFICIENT SPARSIFICATION OF DENSELY CONNECTED CLUSTERS

## ABSTRACT

When modelling a real-world dataset as a graph, groups of highly correlated data items correspond to densely connected vertex sets (clusters), and efficient algorithms that find these clusters have broad applications in various data analysis tasks. In this paper we study densely connected clusters in graphs and introduce two sparsification algorithms that preserve the structure of these clusters in both undirected graphs and directed ones. We show that our algorithms significantly speedup the running time of existing clustering algorithms while preserving their effectiveness.

## 1 INTRODUCTION

Graph clustering is a fundamental technique in data analysis with wide-ranging applications in machine learning and data science. A classical graph clustering problem involves partitioning the vertices of a graph into sets of highly connected vertices to minimise the normalised cut value. However, many real-world clustering tasks are defined by alternative objective functions, tailored to the specific needs and constraints of the problem at hand. One such example involves uncovering the vertex sets (clusters) that are densely connected to each other, and these clusters are connected through bipartite-like graphs. For example, when representing the migration or trade datasets with a graph, a pair of densely connected clusters captures regional migration or trade patterns (Cucuringu et al., 2020; Laenen & Sun, 2020; He et al., 2022), and the importance of these densely connected clusters extends to various other real-world datasets (Bennett et al., 2022; Concas et al., 2022).

In this paper we study densely connected clusters in both undirected graphs and directed ones. We first study the case for undirected graphs, and present an efficient algorithm that sparsifies an input graph while preserving its densely connected clusters. For any undirected $G = (V, E)$ and a pair of disjoint non-empty subsets $V_1, V_2 \subset V$, let $\overline{\phi}_G(V_1, V_2)$ be

$$\overline{\phi}_G(V_1, V_2) \triangleq \frac{2w_G(V_1, V_2)}{\mathrm{vol}_G(V_1 \cup V_2)},$$

and for every $k \in \mathbb{N}$ we define the $k$-way dual Cheeger constant by

$$\bar{\rho}_G(k) \triangleq \max_{(A_1,B_1),\ldots,(A_k,B_k)} \min_{1 \leq i \leq k} \overline{\phi}_G(A_i, B_i), \tag{1.1}$$

where the maximum is taken over all the possible $k$ pairs of subsets $(A_1, B_1), \ldots, (A_k, B_k)$ satisfying $A_i \cap A_j = \emptyset, B_i \cap B_j = \emptyset, A_i \cap B_j = \emptyset$ for different $i, j \in [k]$, and $A_i \cup B_i \neq \emptyset$ for different $i, j \in [k]$. Notice that a high value of $\bar{\rho}_G(k)$ implies that $G$ contains $k$ disjoint pairs of densely connected $(A_i, B_i)$'s, i.e., almost-bipartite components. We prove that, when $G$ presents a clear structure of exactly $k$ pairs of densely connected clusters with respect to $\bar{\rho}_G(k)$, this structure can be represented by a sparse subgraph $H$ of $G$ with $\widetilde{O}(n)$ edges, and $H$ can be constructed in nearly-linear time[1]. Our result is as follows:

**Theorem 1** (Result for undirected graphs)**.** *Let $G = (V_G, E_G, w_G)$ be an undirected and weighted graph of $m$ edges, and assume that $G$ contains $k$ pairs of densely-connected clusters*

---

[1] We say that a graph algorithm runs in nearly-linear time if the algorithm's running time is $O(m \cdot \mathrm{poly} \log n)$, where $m$ and $n$ are the number of edges and vertices of the input graph. For simplicity, we use $\widetilde{O}(\cdot)$ to hide a poly-logarithmic factor of $n$.

$(A_1, B_1), \ldots, (A_k, B_k)$ corresponding to $\bar{\rho}_G(k)$. Then, there is an algorithm that runs in $\widetilde{O}(m)$ time and computes a sparsifier $G^* = (V_G, F \subset E_G, \widetilde{w})$, such that these $k$ pairs of densely-connected clusters of $G$ is preserved in $G^*$ with high probability. That is, it holds with high probability that $\bar{\rho}_{G^*}(k) = \Omega(\bar{\rho}_G(k))$, and $G^*$ contains only $k$ pairs densely-connected clusters.

Secondly, we study the densely connected clusters in directed graphs. Let $\overrightarrow{G} = (V_{\overrightarrow{G}}, E_{\overrightarrow{G}}, w_{\overrightarrow{G}})$ be a digraph with weight function $w_{\overrightarrow{G}} : E_{\overrightarrow{G}} \to \mathbb{R}_{\geq 0}$. For any vertex $u \in V_{\overrightarrow{G}}$, we use $\deg_{\text{out}}(u)$ or $\deg_{\text{in}}(u)$ to denote the sum of weights of directed edges with $u$ as the tail or the head, respectively. For any $S \subset V_{\overrightarrow{G}}$, we define $\text{vol}_{\text{out}}(S) = \sum_{u \in S} \deg_{\text{out}}(u)$ and $\text{vol}_{\text{in}}(S) = \sum_{u \in S} \deg_{\text{in}}(u)$. For any two disjoint subsets $A, B \subset V_{\overrightarrow{G}}$, we define $\overline{\phi}_{\overrightarrow{G}}(A, B)$ by

$$\overline{\phi}_{\overrightarrow{G}}(A, B) \triangleq \frac{2w_{\overrightarrow{G}}(A, B)}{\text{vol}_{\text{out}}(A) + \text{vol}_{\text{in}}(B)}, \tag{1.2}$$

where $w_{\overrightarrow{G}}(A, B)$ is the sum of the weights of the edges from $A$ to $B$. For every $k \in \mathbb{N}$, the $k$-way directed dual Cheeger constant is defined by

$$\bar{\rho}_{\overrightarrow{G}}(k) \triangleq \max_{(A_1, B_1), \ldots, (A_k, B_k)} \min_{1 \leq i \leq k} \overline{\phi}_{\overrightarrow{G}}(A_i, B_i), \tag{1.3}$$

where the maximum is taken over all the possible $k$ pairs of subsets $(A_1, B_1), \ldots, (A_k, B_k)$ satisfying $A_i \cap A_j = \emptyset, B_i \cap B_j = \emptyset, A_i \cap B_j \neq \emptyset$ for different $i, j \in [k]$, $A_i \cup B_i \neq \emptyset$ for any $i \in [k]$. By definition, a high value of of $\bar{\rho}_{\overrightarrow{G}}(k)$ implies that graph $\overrightarrow{G}$ contains $k$ pairs of clusters $(A_1, B_1), \ldots, (A_k, B_k)$ such that almost all edges with their tails in $A_i$ have their head in $B_i$ and conversely almost all edges with their head in $B_i$ have their tail in $A_i$. We prove that, when $\overrightarrow{G}$ presents a structure of $k$ densely connected clusters with respect to $\bar{\rho}_{\overrightarrow{G}}(k)$, this structure is preserved in graph $\overrightarrow{G^*}$ with $\widetilde{O}(n)$ edges, and $\overrightarrow{G^*}$ can be constructed in nearly-linear time. Our result is as follows:

**Theorem 2** (Result for directed graphs). *Let $\overrightarrow{G} = (V_{\overrightarrow{G}}, E_{\overrightarrow{G}}, w_{\overrightarrow{G}})$ be a directed and weighted graph of $m$ edges, and assume that $\overrightarrow{G}$ contains $k$ pairs of densely-connected clusters $(A_1, B_1), \ldots, (A_k, B_k)$ with respect to $\bar{\rho}_{\overrightarrow{G}}(k)$. Then, there is an algorithm that runs in $\widetilde{O}(m)$ time and computes a sparsifier $\overrightarrow{G^*} = (V_{\overrightarrow{G}}, F \subset E_{\overrightarrow{G}}, \widetilde{w})$, such that these $k$ pairs of densely-connected clusters of $\overrightarrow{G}$ are preserved in $\overrightarrow{G^*}$ with high probability. That is, it holds with high probability that $\bar{\rho}_{\overrightarrow{G^*}}(k) = \Omega(\bar{\rho}_{\overrightarrow{G}}(k))$, and $\overrightarrow{G^*}$ only contains $k$ pairs of densely-connected clusters.*

To examine the significance of Theorems 1 and 2, we first highlight that our algorithms preserve the cut values $w(A_i, B_i)$ between the pairs of vertex sets $A_i$ and $B_i$ for $1 \leq i \leq k$; this objective is very different from the one for most graph sparsification problems, which only preserve the cut values between vertex set $S$ and $V \setminus S$. Secondly, our algorithms preserve $k$ pairs of densely connected clusters, and the value of $k$ in the output graph is the same as the original input graph. Thirdly, our second result works for directed graphs; this result is very interesting on its own since most sparsification algorithms are only applicable for undirected graphs.

The design of our algorithms is based on several reductions and sampling routines that can be implemented *locally* when the degree sequence of the underlying graph is available with an oracle. As such one can run our algorithms *online* while exploring the underlying graph with existing local algorithms that find densely connected clusters (e.g., (Andersen, 2010; Li & Peng, 2013)), resulting in direct improvement on the running time of the existing algorithms. To demonstrate this, we conduct experimental studies and show that our algorithms can be directly applied to significantly speed up the running times to the ones presented in (Macgregor & Sun, 2021), while preserving similar output results.

**Related Work.** Trevisan (2009) developed a spectral algorithm that finds two densely connected clusters in an undirected graph, and used this to design an approximation algorithm for the max-cut problem. Li & Peng (2013) and Macgregor & Sun (2021) presented local algorithms that find a pair of densely connected clusters. Cucuringu et al. (2020) proved that densely connected clusters in a digraph can be uncovered through spectral clustering on a complex-valued Hermitian matrix

representation of directed graphs. Neumann & Peng (2022) further presented a sublinear-time oracle which, under a certain condition, correctly classified the membership most vertices in a set of hidden planted ground-truth clusters in signed graphs.

Our work relates to the problem of finding clusters in *disassortative* networks (Moore et al., 2011; Pei et al., 2019; Zhu et al., 2020), although most existing techniques are based on semi-supervised and global methods. Our work is further related to a number of graph sparsification algorithms, e.g., (Spielman & Teng, 2011; Batson et al., 2012; Cohen et al., 2017; Lee & Sun, 2017). In comparison with these results, our algorithms are much easier to implement, and work for directed graphs.

## 2 PRELIMINARIES

In this section we list the notation and background knowledge of spectral graph theory.

**Matrix Representation of Graphs.** We always use $G = (V, E, w)$ to represent an undirected and weighted graph with $n$ vertices and weight function $w : E \to \mathbb{R}_{\geq 0}$. The degree of any vertex $u$ is defined as $d_G(u) = \sum_{u \sim v} w(u, v)$, where the notation $u \sim v$ represents that $u$ and $v$ are adjacent, i.e., $\{u, v\} \in E(G)$. For any set $S \subset V$ in $G$, the volume of $S$ is defined by $\mathrm{vol}_G(S) = \sum_{u \in S} d_G(u)$. The normalised indicator vector of a set $S \subset V$ is defined by $\chi_S(v) = \sqrt{\frac{d_G(v)}{\mathrm{vol}_G(S)}}$ if $v \in S$, and $\chi_S(v) = 0$ otherwise. Let $A_G$ be the adjacency matrix of $G$ defined by $(A_G)_{u,v} = w(u, v)$ if $\{u, v\} \in E(G)$, and $(A_G)_{u,v} = 0$ otherwise. The degree matrix $D_G$ of $G$ is a diagonal matrix defined by $(D_G)_{u,u} = d_G(u)$, and the normalised Laplacian of $G$ is defined by $\mathcal{L}_G = I - D_G^{-1/2} A_G D_G^{-1/2}$. We can also write the normalised Laplacian matrix with respect to the indicator vectors of the vertices: for each vertex $v$, we define an indicator vector $\chi_v \in \mathbb{R}^n$ by $\chi_v(u) = \frac{1}{\sqrt{d_v}}$ if $u = v$, and $\chi_v(u) = 0$ otherwise. We further define $b_e = \chi_u - \chi_v$ for each edge $e = \{u, v\}$, where the orientation of $e$ is chosen arbitrarily. Then, we have $\mathcal{L}_G = \sum_{e=\{u,v\} \in E} w(u, v) \cdot b_e b_e^\mathsf{T}$. We also define

$$\mathcal{J}_G \triangleq I + D_G^{-1/2} A_G D_G^{-1/2}.$$

For any symmetric matrix $A \in \mathbb{R}^{n \times n}$, we use $\lambda_1(A) \leq \lambda_2(A) \leq \cdots \leq \lambda_n(A)$ to express the eigenvalues of $A$. For ease of presentation, we always use $0 = \lambda_1 \leq \lambda_2 \leq \cdots \leq \lambda_n \leq 2$ to express the eigenvalues of $\mathcal{L}_G$, with the corresponding orthonormal eigenvectors $f_1, f_2, \cdots, f_n$. With slight abuse of notation, we use $\mathcal{L}_G^{-1}$ for the pseudo inverse of $\mathcal{L}_G$, i.e.,

$$\mathcal{L}_G^{-1} \triangleq \sum_{i=2}^{n} \frac{1}{\lambda_i} f_i f_i^\mathsf{T}.$$

Note that when $G$ is connected, it holds that $\lambda_2 > 0$ and the matrix $\mathcal{L}_G^{-1}$ is well defined. We sometimes drop the subscript $G$ when it is clear from the context.

For any vector $x \in \mathbb{R}^n$ we define $\|x\| \triangleq \sqrt{\sum_{i=1}^{n} x_i^2}$, and any matrix $M \in \mathbb{R}^{n \times n}$ we define

$$\|M\| = \max_{x \in \mathbb{R}^n \setminus \{\mathbf{0}\}} \frac{\|Mx\|}{\|x\|}.$$

**Graph expansion and Cheeger inequality.** For any undirected graph $G$, the expansion (or conductance) of any non-empty subset $S \subset V$ in $G$ is defined as $\phi_G(S) \triangleq \frac{w_G(S, \bar{S})}{\mathrm{vol}_G(S)}$, where $\bar{S}$ is the complement of $S$ and $w_G(S, \bar{S}) = \sum_{u \in S, v \in \bar{S}} w_G(u, v)$. We call subsets of vertices $S_1, S_2, \cdots, S_k$ a $k$-way partition of $G$ if $S_i \neq \emptyset$ for all $1 \leq i \leq k$, $S_i \cap S_j = \emptyset$ for $i \neq j$ and $\bigcup_{i=1}^{k} S_i = V$. For every $k \in \mathbb{N}$, the $k$-way expansion constant is defined as

$$\rho_G(k) = \min_{S_1, S_2, \cdots, S_k} \max_{1 \leq i \leq k} \phi_G(S_i),$$

where the minimum is taken over all possible $k$-way partitions of $G$. Lee et al. (2014) proves the following higher-order Cheeger inequality:

**Lemma 3** (Higher-order Cheeger Inequality, (Lee et al., 2014)). *It holds for any undirected graph $G$ of $n$ vertices and integer $1 \leq k \leq n$ that $\lambda_k/2 \leq \rho_G(k) \leq Ck^2\sqrt{\lambda_k}$, where $C$ is a universal constant.*

Generalising this, Liu (2015) proves the following higher-order dual-Cheeger inequality:

**Lemma 4** (Higher-order dual-Cheeger Inequality, (Liu, 2015)). *It holds for any undirected graph $G$ of $n$ vertices and integer $1 \leq k \leq n$ that $(2 - \lambda_{n-k+1})/2 \leq 1 - \bar{\rho}_G(k) \leq Ck^3\sqrt{2 - \lambda_{n-k+1}}$, where $C$ is a universal constant.*

Note that the higher-order dual Cheeger inequality can be viewed as a quantitative version of the fact that $\lambda_{n-k+1} = 2$ if and only if $G$ has at least $k$ bipartite connected components.

## 3 SPARSIFYING DENSELY CONNECTED CLUSTERS IN UNDIRECTED GRAPHS

In this section we present a nearly-linear time sparsification algorithm such that every pair of densely connected clusters in an undirected graph $G$ is approximately preserved in the sparsifed graph $G^*$, and sketch the proof. Our result is as follows:

**Theorem 5** (Formal Statement of Theorem 1). *There exists an algorithm that, given a graph $G = (V, E, w)$ with $\bar{\rho}_G(k) \geq \frac{1}{\log n}$ for constant some $k$ as input, with high probability computes a sparsifier $G^* = (V, F \subset E, \widetilde{w})$ with $|F| = O\left(\frac{n \cdot \log^3 n}{2 - \lambda_{n-k}}\right)$ edges such that the following hold: (1) it holds that $\bar{\rho}_{G^*}(k) = \Omega(\bar{\rho}_G(k))$; (2) it holds that $\lambda_{k+1}(\mathcal{J}_{G^*}) = \Theta(\lambda_{k+1}(\mathcal{J}_G))$.*

The first statement of Theorem 5 shows that the $k$ pairs of densely connected clusters of $G$ is approximately preserved in $G^*$, and together with Lemma 4 the second statement shows that the number of pairs of the densely connected clusters in $G$ and $G^*$ is the same.

**Algorithm.** Our algorithm is similar with Sun & Zanetti (2019) at a high level, and is based on sampling edges in $G$ with certain probabilities. Formally, for an input undirected graph $G = (V, E, w_G)$, the algorithm starts with $G^* = (V, \emptyset, \widetilde{w})$ and samples every edge $u \sim v$ in $G$ with probability

$$p_e \triangleq p_u(v) + p_v(u) - p_u(v) \cdot p_v(u),$$

where

$$p_u(v) \triangleq \min\left\{w_G(u, v) \cdot \frac{C \cdot \log^3 n}{d_G(u) \cdot (2 - \lambda_{n-k})}, 1\right\}. \tag{3.1}$$

For every sampled edge $e = \{u, v\}$, the algorithm adds $e$ to graph $G^*$, and sets $w_{G^*}(e) = w_G(e)/p_e$.

**Proof Sketch of Theorem 5.** We first show prove that the cut values between $A_i$ and $B_i$ in $G$ is preserved in $H$ for any $1 \leq i \leq k$. For any edge $e = \{u, v\}$, we define the random variable $Y_e$ by $Y_e = w_G(u, v)/p_e$ with probability $p_e$, and $Y_e = 0$ otherwise. By defining $X = w_H(A_i, B_i)$, we prove that $\mathbf{E}[X] = w_G(A_i, B_i)$ and

$$\mathbf{E}\left[X^2\right] \leq \frac{2 - \lambda_{n-k}}{C \cdot \log^3 n} \sum_{\substack{e=\{u,v\} \\ u \in A_i, v \in B_i}} w(u, v) \cdot \left(\frac{d_G(u) + d_G(v)}{2}\right).$$

Let $\{(A_i, B_i)\}_{i=1}^k$ be the optimal cluster where $\bar{\rho}(k)$ is attained for graph $G$. Then, we have for every $1 \leq i \leq k$ that

$$\bar{\rho}_G(k) \leq \overline{\phi}_G(A_i, B_i) = \frac{2w_G(A_i, B_i)}{\text{vol}_G(A_i \cup B_i)},$$

which implies

$$\frac{\bar{\rho}_G(k)}{2} \cdot \text{vol}_G(A_i \cup B_i) \leq \sum_{\substack{e=\{u,v\} \\ u \in A_i, v \in B_i}} w_G(u, v).$$

Applying the Chebyshev's inequality, we have for any constant $c \in \mathbb{R}^+$ that

$$\mathbf{P}\left[|X - \mathbf{E}[X]| \geq c \cdot \mathbf{E}[X]\right] \leq \frac{\mathbf{E}[X^2]}{c^2 \cdot \mathbf{E}[X]^2}$$

$$\leq \frac{2 \cdot (2 - \lambda_{n-k})}{c^2 \cdot C \cdot \log^3 n \cdot \bar{\rho}_G(k)^2} \cdot \frac{\left(\max_{\substack{e = \{u,v\} \\ u \in A_i, v \in B_i}} \{d_G(u) + d_G(v)\}\right) \cdot \sum_{\substack{e = \{u,v\} \\ u \in A_i, v \in B_i}} w_G(u,v)}{\text{vol}_G(A_i \cup B_i)^2}.$$

Since $\text{vol}_G(A_i \cup B_i) = \sum_{u \in A_i} d_G(u) + \sum_{v \in B_i} d_G(v)$ and $d_G(u) = \sum_{u \sim v} w_G(u,v)$, we have

$$\max_{\substack{e = \{u,v\} \\ u \in A_i, v \in B_i}} \{d_G(u) + d_G(v)\} \leq \sum_{u \in A_i} d_G(u) + \sum_{v \in B_i} d_G(v) = \text{vol}_G(A_i \cup B_i)$$

and

$$\sum_{\substack{e = \{u,v\} \\ u \in A_i, v \in B_i}} w_G(u,v) \leq \text{vol}_G(A_i \cup B_i).$$

Applying these gives us that

$$\mathbf{P}\left[|X - \mathbf{E}[X]| \geq c \cdot \mathbf{E}[X]\right] \leq \frac{2(2 - \lambda_{n-k})}{c^2 \cdot C \cdot \log^3 n \cdot \bar{\rho}(k)^2} = O\left(\frac{1}{\log n}\right).$$

Hence, by the union bound, we have that $w_H(A_i, B_i) = \Omega\left(w_G(A_i, B_i)\right)$ for all $1 \leq i \leq k$. The second statement of Theorem 5 holds by the analysis similar with Sun & Zanetti (2019). Finally, the total number of edges in $H$ follows by the definition of sampling probability and the Markov inequality. This completes the proof of Theorem 5.

# 4 SPARSIFYING DENSELY CONNECTED CLUSTERS IN DIRECTED GRAPHS

In this section we present a nearly-linear time sparsification algorithm such that all pairs of densely connected clusters in a directed graph is approximately preserved in the output sparsifier, and prove Theorem 2. Specifically, for a digraph $\overrightarrow{G}$ that contains exactly $k$ pairs of $(A_1, B_1), \ldots, (A_k, B_k)$ with high values of $\overrightarrow{\phi}_{\overrightarrow{G}}(A_i, B_i)$ for every $1 \leq i \leq k$, our objective is to construct a sparse digraph $\overrightarrow{G^*}$, such that (i) the values of $\overrightarrow{\phi}_{\overrightarrow{G^*}}(A_i, B_i)$ are high for every $1 \leq i \leq k$ and (ii) the number of such pairs in $\overrightarrow{G^*}$ is the same as $\overrightarrow{G}$.

Before sketching our technique, we recall that, for undirected graphs, the value of $k$ is proven to be identical for $G$ and $G^*$ by analysing the eigenvalues of $\mathcal{J}_G$ and $\mathcal{J}_{G^*}$ and applying the higher-order dual-Cheeger inequality (Lemma 4). However, a natural matrix representation for directed graph could result in complex-valued eigenvalues, and there is no analog of Lemma 4 for directed graphs. To overcome this, our developed algorithm is based on a reduction from a directed graph to an undirected one, and its reverse operation. Specifically, our designed algorithm consists of the following three steps, as illustrated in Figure 1:

1. for any input digraph $\overrightarrow{G}$, the algorithm constructs an undirected graph $H$ such that every two densely connected clusters $(A_i, B_i)$ in $\overrightarrow{G}$ corresponds to a low-conductance set in $H$;

2. the algorithm constructs a sparsifier $H^*$ of $H$, such that $H$ and $H^*$ have the same structure of clusters;

3. the algorithm applies the sparsified undirected graph $H^*$ to construct a directed graph $\overrightarrow{G^*}$ of $\overrightarrow{G}$ that satisfies $\bar{\rho}_{\overrightarrow{G^*}}(k) = \Omega\left(\bar{\rho}_{\overrightarrow{G}}(k)\right)$.

**Constructing $H$ from $\overrightarrow{G}$.** Notice that, to preserve $\overrightarrow{\phi}_{\overrightarrow{G^*}}(A_i, B_i)$, the cut values $w(A_i, B_i)$ between $A_i$ and $B_i$ need to be approximately preserved in a sparsified directed graph; this objective is very different from the most graph sparsification one, which only preserves the cut value between any set $S$ and its complement. To overcome this, following (Macgregor & Sun, 2021) we construct an

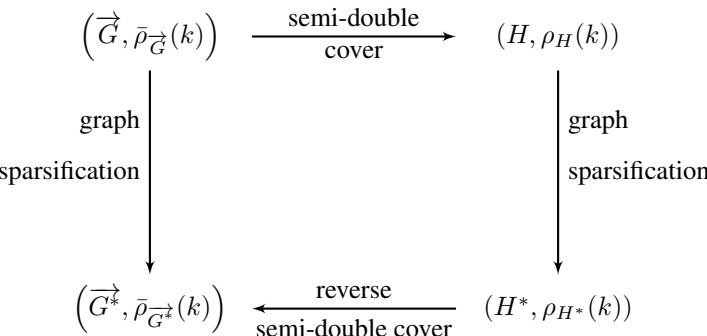

Figure 1: A commutative diagram of sparsification of directed graphs. In order to construct $\overrightarrow{G^*}$ from $\overrightarrow{G}$, we construct graphs $H$ and $H^*$ and prove the close relationships between $\overrightarrow{G}$, $H$, $H^*$, and $\overrightarrow{G^*}$.

undirected graph $H$ such that every pair of densely connected clusters $(A_i, B_i)$ in $\overrightarrow{G}$ corresponds to a low-conductance set in $H$. Specifically, for a weighted digraph $\overrightarrow{G} = (V_{\overrightarrow{G}}, E_{\overrightarrow{G}}, w_{\overrightarrow{G}})$, we construct its semi-double cover $H = (V_H, E_H, w_H)$ as follows:

1. every vertex $v \in V_{\overrightarrow{G}}$ has two corresponding vertices $v_1, v_2 \in V_H$;
2. for every edge $u \to v \in E_{\overrightarrow{G}}$, we add the edge $\{u_1, v_2\}$ in $E_H$.

See Figure 2 for illustration.

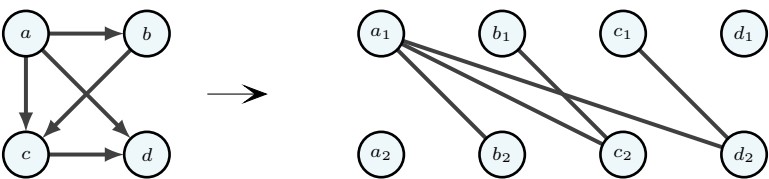

Figure 2: Illustration of the semi-double cover construction. A directed graph of $n$ vertices (left) corresponds to an undirected and bipartite graph of $2n$ vertices (right).

Next we analyse the properties of the reduced graph. Let $\overrightarrow{G}$ be a digraph with semi-double cover $H$. For any $S \subset V_{\overrightarrow{G}}$, we define $S_1 \subset V_H$ and $S_2 \subset V_H$ by $S_1 \triangleq \{v_1 | v \in S\}$ and $S_2 \triangleq \{v_2 | v \in S\}$. A subset $S$ of $V_H$ is called *simple* if $|\{v_1, v_2\} \cap S| \leq 1$ holds for all $v \in V_{\overrightarrow{G}}$. The following lemma develops a relationship between the *flow ratio* from $A$ to $B$ defined by

$$f_{\overrightarrow{G}}(A, B) \triangleq 1 - \overline{\phi}_{\overrightarrow{G}}(A, B) \tag{4.1}$$

and $\Phi_H(A_1 \cup B_2)$, for any $A, B$.

**Lemma 6.** *Let $\overrightarrow{G}$ be a digraph with semi-double cover $H$. Then, it holds for any $A, B \subset V_{\overrightarrow{G}}$ that $f_{\overrightarrow{G}}(A, B) = \phi_H(A_1 \cup B_2)$. Similarly, for any simple set $S \subset V_H$, let $A = \{u : u_1 \in S\}$ and $B = \{u : u_2 \in S\}$. Then, it holds that $f_{\overrightarrow{G}}(A, B) = \phi_H(S)$.*

Lemma 6 proves a one-to-one correspondence between any pair of disjoint vertex sets in $\overrightarrow{G}$ and a vertex set in $H$. Building on this, we prove that this one-to-one correspondence can be generalised between any $k$ pairs of densely connected clusters in $\overrightarrow{G}$ and $k$ disjoint vertex sets in $H$. Moreover, the structure of $k$ pairs of densely connected clusters in $\overrightarrow{G}$ is preserved by a collection of $k$ disjoint vertex sets of low conductance in $H$.

**Lemma 7.** *For any directed and weighted graph $\overrightarrow{G} = (V_{\overrightarrow{G}}, E_{\overrightarrow{G}}, w_{\overrightarrow{G}})$ and $k \in \mathbb{N}$, it holds that*

$$\bar{\rho}_{\overrightarrow{G}}(k) = 1 - \min_{C_1, \ldots, C_k} \max_{1 \leq i \leq k} \phi_H(C_i), \tag{4.2}$$

*where the minimum is taken over a collection of $k$ disjoint simple subsets of $V_H$ defined by $C_i = A_{i_1} \cup B_{i_2}$ for $1 \leq i \leq k$.*

**Sparsification of $H$.** Next we construct a sparse representation of $H$, denoted by $H^*$, such that the $k$ vertex sets of low conductance is preserved in $H^*$. To achieve this, we apply the following result to construct a cluster-preserving sparsifier, which guarantees that the structure of $k$ clusters in $G$ is preserved in $H$.

**Lemma 8** ((Sun & Zanetti, 2019)). *There exists an algorithm that, given a graph $G = (V, E, w)$ with $k$ clusters as input, with probability at least $9/10$, computes a sparsifier $H = (V, F \subset E, \widetilde{w})$ with $|F| = O(1/\lambda_{k+1} \cdot n \log n)$ edges such that the following holds:*

1. *it holds for any $1 \leq i \leq k$ that $\phi_H(S_i) = O(k \cdot \phi_G(S_i))$, where $S_1, \cdots, S_k$ are the optimal clusters in $G$ that achieves $\rho(k)$.*

2. $\lambda_{k+1}(\mathcal{L}_H) = \Omega(\lambda_{k+1}(\mathcal{L}_G))$.

**Constructing $\overrightarrow{G^*}$ from $H^*$.** Finally, we construct a directed graph $\overrightarrow{G^*}$ from $H^*$ such that the original $k$ pairs of densely connected clusters in $\overrightarrow{G}$ is preserved in $\overrightarrow{G^*}$. To achieve this, we introduce the following *reverse semi-double cover*:

**Definition 9** (reverse semi-double cover). *Given any double cover graph $H^* = (V_{H^*}, E_{H^*}, w_{H^*})$ as input, the reverse semi-double cover of $H^*$ is a directed graph $\overrightarrow{G^*} = (V_{\overrightarrow{G^*}}, E_{\overrightarrow{G^*}}, w_{\overrightarrow{G^*}})$ constructed as follows:*

- *every pair of vertices $u_1$ and $u_2$ in $V_{H^*}$ corresponds to a vertex $v \in V_{\overrightarrow{G^*}}$;*

- *we add an edge $u \to v$ to $E_{\overrightarrow{G}}$ if there is edge $\{u_1, v_2\} \in E_{H^*}$, and set $w_{\overrightarrow{G^*}}(u, v) = w_{H^*}(u_1, v_2)$.*

One might think that the reverse double cover plays an exact opposite role of the double cover, however it is not the case. In particular, while our constructed subsets $C_1, \ldots, C_k$ in the first step are always simple in $H$ (cf. Lemma 7), the $k$ subsets corresponding to $\rho_H(k)$ are not necessarily simple. As a result,

$$\min_{C_1, \ldots, C_k} \max_{1 \leq i \leq k} \phi_H(C_i) = \rho_H(k)$$

doesn't hold in general, and there is no direct correspondence between $C_1, \ldots, C_k$ in $H$ and the $k$ pairs of densely connected clusters in $\overrightarrow{G^*}$ that correspond to $\bar{\rho}_{\overrightarrow{G^*}}(k)$.

To analyse $\rho_{\overrightarrow{G^*}}(k)$, for any set $S \subset V_H$ we partition the set into two subsets $S_1$ and $S_2$ defined by $S_1 = S \cap (A_{i_1} \cup B_{i_2})$ and $S_2 = S \cap (A_{i_2} \cup B_{i_1})$. For example, following Figure 2, if the sets $A_i = \{a, c\}$ and $B_i = \{b, d\}$ and the set $S \subset V_H$ be $S = \{a_1, b_1, b_2, c_1, c_2\}$, then we have $S_1 = \{a_1, b_2, c_1\}$ and $S_2 = \{b_1, c_2\}$. Since $A_i$ and $B_i$ are densely connected in $H$, implying that most of the edges are either from $A_i$ to $B_i$ or from $B_i$ to $A_i$, there are few edges within $A_i$ and $B_i$ for $1 \leq i \leq k$. Hence, there are very few edges between $S_1$ and $S_2$ for any $S \subset V_H$. Without loss of generality, we assume that

$$\frac{2w_H(S_1, S_2)}{w_H(S_1, \bar{S}_1) + w_H(S_2, \bar{S}_2)} \leq c$$

for some constant $c < 1$. Simplifying the inequality above we get

$$w_H(S_1, \bar{S}_1) + w_H(S_2, \bar{S}_2) - 2w_H(S_1, S_2) \geq (1 - c) \cdot \left[ w_H(S_1, \bar{S}_1) + w_H(S_2, \bar{S}_2) \right].$$

Thus, for any set $S \subset V_H$ that is not necessarily simple we have

$$\phi_H(S) = \frac{w_H(S, \bar{S})}{\text{vol}(S)} = \frac{w_H(S_1, \bar{S}_1) + w_H(S_2, \bar{S}) - 2w_H(S_1, S_2)}{\text{vol}(S_1) + \text{vol}(S_2)}$$

$$\geq (1 - c) \cdot \min \left\{ \frac{w_H(S_1, \bar{S}_1)}{\text{vol}(S_1)}, \frac{w_H(S_2, \bar{S}_2)}{\text{vol}(S_2)} \right\} = (1 - c) \cdot \min \left\{ \phi_H(S_1), \phi_H(S_2) \right\},$$

where the last inequality follows from the median inequality. Thus, for every set $S \subset V_H$, there exists a simple set $T \subset V_H$ such that $\phi_H(S) \geq (1 - c) \cdot \phi_H(T)$. Moreover, for any collection

of $k$-disjoint sets $S_1, S_2, \cdots, S_k$, where $S_i \subset V_H$ we have a collection of $k$-disjoint simple sets $T_1, T_2, \cdots, T_k$, where $T_i \subset V_H$, such that

$$\max_{1 \leq i \leq k} \phi_H(S_i) \geq (1 - c) \cdot \max_{1 \leq i \leq k} \phi_H(T_i).$$

Taking minimum over all such collection of $k$-disjoint subsets of $V_H$ gives us that

$$\min_{S_1, S_2, \cdots, S_k} \max_{1 \leq i \leq k} \phi_H(S_i) = \rho_H(k) \geq (1 - c) \cdot \min_{T_1, T_2, \cdots, T_k} \max_{1 \leq i \leq k} \phi_H(T_i),$$

where in the second half of the inequality the minimum is taken over collection of $k$-disjoint simple subsets of $V_H$. On one hand, rearranging the above inequality we have

$$\frac{1}{1 - c} \cdot \rho_H(k) \geq \min_{T_1, T_2, \cdots, T_k} \max_{1 \leq i \leq k} \phi_H(T_i), \tag{4.3}$$

and on the other hand, since the collection of $k$-disjoint simple subsets of $V_H$ is a sub-collection of the collection of $k$-disjoint subsets of $V_H$, we have

$$\min_{T_1, T_2, \cdots, T_k} \max_{1 \leq i \leq k} \phi_H(T_i) \geq \rho_H(k). \tag{4.4}$$

Thus, combining (4.3) and (4.4), we have

$$\frac{1}{1 - c} \cdot \rho_H(k) \geq \min_{T_1, T_2, \cdots, T_k} \max_{1 \leq i \leq k} \phi_H(T_i) \geq \rho_H(k). \tag{4.5}$$

Further, combining (4.2) and (4.5) we have

$$1 - \frac{1}{1 - c} \cdot \rho_H(k) \leq \bar{\rho}_{\overrightarrow{G}}(k) \leq 1 - \rho_H(k). \tag{4.6}$$

**Proof of Theorem 2.** Now we are ready to prove 2. Since $\overrightarrow{G}$ is a directed graph with $k$ pairs of densely connected clusters, the value of $\bar{\rho}_{\overrightarrow{G}}(k)$ is high; together with (4.6), this implies that $\rho_H(k) = o(1)$. By Lemma 8, we know that there exists a sparsifier $H^*$ of $H$, such that $\rho_{H^*}(k) = O(k \cdot \rho_H(k))$. Thus, we can conclude that $\rho_{H^*}(k) = o(1)$. Hence, applying (4.6) for $\overrightarrow{G^*}$ and $H^*$ we have

$$1 - \frac{1}{1 - c} \cdot \rho_{H^*}(k) \leq \bar{\rho}_{\overrightarrow{G^*}}(k) \leq 1 - \rho_{H^*}(k). \tag{4.7}$$

Finally, using the fact that $\rho_{H^*}(k) = o(1)$, we conclude that $\bar{\rho}_{\overrightarrow{G^*}}(k)$ is close to 1 and hence the structure of $\overrightarrow{G}$ will be preserved in $\overrightarrow{G^*}$. Moreover, by the construction of $H$, and $H^*$, and $\overrightarrow{G^*}$, the value of $k$ is preserved.

For the running time, notice that all the intermediate graphs $H$ and $H^*$ can be constructed locally, and therefore it's sufficient to examine every edge of the input graph $\overrightarrow{G}$ once throughout the execution of the algorithm. This implies the nearly-linear running time of our overall algorithm. Combining everything above above proves Theorem 2.

## 5 EXPERIMENTS

In this section, we evaluate the performance of our proposed algorithms on synthetic data sets. We employ the algorithms presented in (Macgregor & Sun, 2021) as the baseline algorithms, and examine the speedup of their algorithms when applying our sparsification algorithms as subroutines. Notice that, as all the involved operations of our algorithms can be performed locally, one can run our graph sparsification algorithms online while exploring the underlying graph with a local algorithm. For ease of presentation, in this section we call the local algorithm in (Macgregor & Sun, 2021) with our sparsification framework our algorithm. All experiments were performed on a HP ZBook Studio with 11th Gen Intel(R) Core(TM) i7-11800H @ 2.30GHz processor and 32 GB of RAM.

## 5.1 RESULTS FOR UNDIRECTED GRAPHS

We compare the performance of our algorithm with the previous existing algorithm LocBipartDC given by Macgregor & Sun (2021), which we refer to as MS, on synthetic graphs generated from the stochastic block model (SBM). Specifically, we assume that the graph has $k = 2$ clusters, say $C_1, C_2$, and the number of vertices in each cluster, denoted by $n_1$ and $n_2$ respectively, satisfies $n_1 = n_2$. Moreover, any pair of vertices $u \in C_i$ and $v \in C_j$ is connected with probability $p_{ij}$. We assume that $p_{12} = p_{21} = p$ and $p_{11} = p_{22} = q$, where $q = 0.1p$. Throughout the experiments, we leave the parameters $n$ and $p$ free but maintain the above relations.

Our algorithm sparsifies the graph in an online manner while exploring it and simultaneously apply the MS algorithm. We evaluate the quality of the output $(L, R)$ returned by each algorithm with respect to its bipartiteness ratio defined by $\beta(L, R) = 1 - \overline{\phi}(L, R)$. All our reported results are the average performance of each algorithm over 10 runs, in which a random vertex from $C_1 \cup C_2$ is chosen as the starting vertex of the algorithm. We generate graphs from the SBM such that $q = 0.1p$ and vary the size of the target set by varying $n_1$ between $1,000$ and $6,000$. In Figure 3, we fix the probability $p = 0.3$ and vary the number of vertices $n_1 = n_2$ and compare both runtime and the bipartiteness ratio between the MS algorithm and our algorithm. One can observe that for a fixed probability $p$ as we increase the number of vertices, our algorithm takes much less time than the MS algorithm and maintains a similar bipartiteness ratio with the MS algorithm.

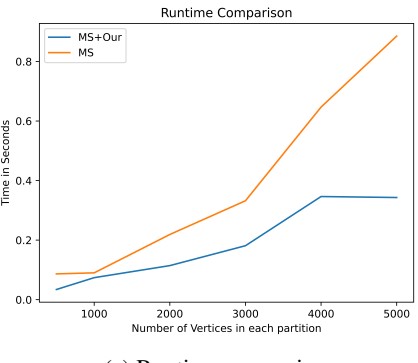

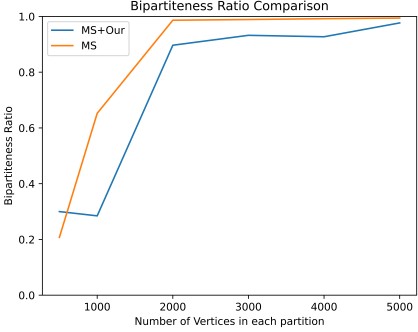

(a) Runtime comparison        (b) Bipartiteness Ratio comparison

Figure 3: Runtime and bipartiteness comparison between MS and our algorithm by fixing $p = 0.3$, $q = 0.1p$ and varying the number of vertices between $500$ and $6,000$.

## 5.2 RESULTS FOR DIRECTED GRAPHS

Next we evaluate the performance of our algorithm for digraphs on synthetic dataset. We compare the performance of our algorithm with the previous existing algorithm EvoCutDirected given by Macgregor & Sun (2021), which we refer to as ECD, and use the graphs generated from the SBM as the algorithms' input. In our algorithm, given a digraph $G$ as input, we sparsify the graph along with generating the volume-biased ESP on $G's$ semi-double cover $H$. Since the ECD is a local algorithm, we also test our algorithm locally. In this model, we look into a cluster which is almost bipartite with the bipartition being $L$ and $R$. We set the number of vertices in $L$ and $R$ to be $n_1$ and $n_2$ such that $n_1 = n_2$ and the probability of an edge to be as follows

$$\begin{array}{cc} & L \qquad R \\ \begin{array}{c} L \\ R \end{array} & \begin{pmatrix} 9/n_1 & \eta \\ 1 - \eta & 9/n_2 \end{pmatrix}, \end{array}$$

i.e., the probability that there is an edge within the partition is $9/n_1 = 9/n_2$ and so on. Since most of our directed edges are from $L$ to $R$, the value of $\eta$ is high. For our experiments we generate two sets of plots:

- We first fix the value of $\eta = 0.7$ and vary the number of vertices in each partition from $2,000$ to $5,000$, and compare the runtime of the ECD algorithm and our algorithm. One

can observe that our algorithm takes much less time than the ECD algorithm and gives a similar flow-ratio at the same time as we increase the number of vertices.

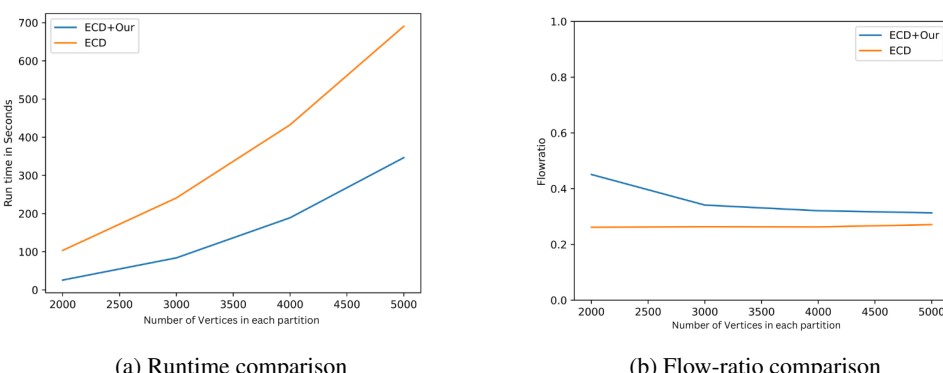

(a) Runtime comparison

(b) Flow-ratio comparison

Figure 4: Runtime and flow-ratio comparison between ECD and our algorithm.

- Based on this, it suffices for us to only compare the running times. We vary the number of vertices in each partition from $1,500$ to $5,000$ and vary the value of $\eta$ from $0.7$ to $0.9$, and compare the runtime of the ECD algorithm and our algorithm. One can observe that our algorithm runs faster than the ECD algorithm as $\eta$ increases, i.e., when the graph is dense.

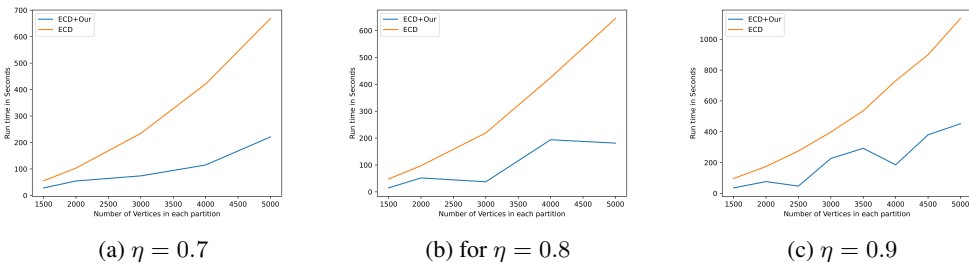

(a) $\eta = 0.7$  (b) for $\eta = 0.8$  (c) $\eta = 0.9$

Figure 5: Runtime comparison between ECD and our algorithm for $\eta = 0.7, 0.8$ and $0.9$.

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

## A USEFUL INEQUALITIES

The following inequalities will be used in our analysis.

**Theorem 10** (Courant-Fischer Theorem). *Let $A$ be a $n \times n$ symmetric matrix with eigenvalues $\lambda_1 \leq \lambda_2 \leq \cdots \leq \lambda_n$. Then, it holds for any $1 \leq k \leq n$ that*

$$\lambda_k = \min_{\substack{S \\ \dim(S)=k}} \max_{y \in S \setminus \{\mathbf{0}\}} \frac{y^{\mathsf{T}} \cdot A \cdot y}{y^{\mathsf{T}} \cdot y} = \max_{\substack{S \\ \dim(S)=n-k+1}} \min_{y \in S \setminus \{\mathbf{0}\}} \frac{y^{\mathsf{T}} \cdot A \cdot y}{y^{\mathsf{T}} \cdot y},$$

*where the maximisation and minimisation are over the subspaces of $\mathbb{R}^n$.*

**Lemma 11** (Bernstein's Inequality). *Let $X_1, X_2, \cdots, X_n$ be independent random variables such that $|X_i| \leq M$ for any $1 \leq i \leq n$. Let $X = \sum_{i=1}^{n} X_i$, and $R = \sum_{i=1}^{n} \mathbf{E}\left[X_i^2\right]$. Then, it holds that*

$$\mathbf{P}\left[|X - \mathbf{E}[X]| \geq t\right] \leq 2\exp\left(-\frac{t^2}{2\left(R + \frac{Mt}{3}\right)}\right).$$

**Lemma 12** (Matrix Chernoff Bound). *Consider a finite sequence $\{X_i\}$ of independent, random, PSD matrices of dimension $d$ that satisfy $\|X_i\| \leq R$. Let $\mu_{\min} = \lambda_{\min}\left(\mathbf{E}[\sum_i X_i]\right)$ and $\mu_{\max} = \lambda_{\max}\left(\mathbf{E}[\sum_i X_i]\right)$. Then, it holds that*

$$\mathbf{P}\left[\lambda_{\min}\left(\sum_i X_i\right) \leq (1-\delta)\mu_{\min}\right] \leq d \cdot \left(\frac{e^{-\delta}}{(1-\delta)^{1-\delta}}\right)^{\frac{\mu_{\min}}{R}} \quad \text{for } \delta \in [0,1],$$

*and*

$$\mathbf{P}\left[\lambda_{\max}\left(\sum_i X_i\right) \geq (1+\delta)\mu_{\max}\right] \leq d \cdot \left(\frac{e^{\delta}}{(1+\delta)^{1+\delta}}\right)^{\frac{\mu_{\max}}{R}} \quad \text{for } \delta \geq 0.$$

## B  OMITTED DETAIL FROM SECTION 3

This section presents all the omitted detail from Section 3, and gives a complete proof of Theorem 5. We first recall that, for every vertex $u$ and its adjacent vertex $v$, the algorithm assigns the edge $e = \{u, v\}$ the probability

$$p_u(v) \triangleq \min\left\{w_G(u,v) \cdot \frac{C \cdot \log^3 n}{d_G(u) \cdot (2 - \lambda_{n-k})}, 1\right\}, \tag{B.1}$$

for a large enough constant $C \in \mathbb{R}_{\geq 0}$. The algorithm checks every edge and samples an edge $e = \{u, v\}$ with probability $p_e$, where

$$p_e \triangleq p_u(v) + p_v(u) - p_u(v) \cdot p_v(u).$$

Note that, it is easy to check that $p_e$ satisfies the inequality

$$\frac{1}{2}(p_u(v) + p_v(u)) \leq p_e \leq p_u(v) + p_v(u).$$

We start with an empty set $F$ and gradually store all the sampled edges in $F$, which is sampled by the algorithm. Finally, the algorithm returns a weighted graph $H = (V, F, w_H)$, where the weight $w_H(u,v)$ of every sampled edge $e = \{u,v\} \in F$ is defined by

$$w_H(u,v) = \frac{w_G(u,v)}{p_e}.$$

Next, we analyze the size of $F$. Since

$$\sum_u \sum_{e=\{u,v\}} w_G(u,v) \cdot \frac{C \cdot \log^3 n}{d_G(u) \cdot (2 - \lambda_{n-k})} = O\left(\frac{n \cdot \log^3 n}{2 - \lambda_{n-k}}\right),$$

it holds by Markov inequality that the number of edges $e = \{u,v\}$ with $p_u(v) \geq 1$ is $O\left(\frac{n \cdot \log^3 n}{2 - \lambda_{n-k}}\right)$. Without loss of generality, we assume that these edges are in $F$, and in the remaining part of the proof we assume it holds for any edge $u \sim v$ that

$$w_G(u,v) \cdot \frac{C \cdot \log^3 n}{d_G(u) \cdot (2 - \lambda_{n-k})} < 1.$$

Then, the expected number of edges in $H$ equals

$$\sum_{e=\{u,v\}} p_e \leq \sum_{e=\{u,v\}} p_u(v) + p_v(u) = \frac{C \cdot \log^3 n}{(2 - \lambda_{n-k})} \sum_{e=\{u,v\}} w(u,v) \cdot \left(\frac{1}{d_G(u)} + \frac{1}{d_G(v)}\right)$$

$$= O\left(\frac{n \cdot \log^3 n}{2 - \lambda_{n-k}}\right),$$

and by Markov inequality it holds with constant probability that

$$|F| = O\left(\frac{n \cdot \log^3 n}{2 - \lambda_{n-k}}\right).$$

Now we show that the cut value between $A_i$ and $B_i$ is preserved in $H$ for all $1 \leq i \leq k$. For any edge $e = \{u, v\}$, we define the random variable $Y_e$ by

$$Y_e = \begin{cases} \dfrac{w_G(u, v)}{p_e} & \text{with probability } p_e, \\ 0 & \text{otherwise.} \end{cases} \tag{B.2}$$

Also, we define $X = w_H(A_i, B_i)$, and have that

$$\mathbf{E}[X] = \sum_{\substack{e=\{u,v\} \\ u \in A_i, v \in B_i}} \mathbf{E}[Y_e] = \sum_{\substack{e=\{u,v\} \\ u \in A_i, v \in B_i}} p_e \cdot \frac{w_G(u, v)}{p_e} = \sum_{\substack{e=\{u,v\} \\ u \in A_i, v \in B_i}} w_G(u, v) = w_G(A_i, B_i). \tag{B.3}$$

Next, we analyse the second moment of the random variable $X$ and have that

$$\begin{aligned}
\mathbf{E}\left[X^2\right] &= \sum_{\substack{e=\{u,v\} \\ u \in A_i, v \in B_i}} p_e \cdot \left(\frac{w_G(u, v)}{p_e}\right)^2 = \sum_{\substack{e=\{u,v\} \\ u \in A_i, v \in B_i}} \frac{w_G(u, v)^2}{p_e} \\
&\leq \sum_{\substack{e=\{u,v\} \\ u \in A_i, v \in B_i}} \frac{2 w_G(u, v)^2}{p_u(v) + p_v(u)} \\
&= \sum_{\substack{e=\{u,v\} \\ u \in A_i, v \in B_i}} \frac{2 w_G(u, v)^2}{\frac{w_G(u,v) \cdot C \cdot \log^3 n}{(2 - \lambda_{n-k})} \cdot \left(\frac{1}{d_G(u)} + \frac{1}{d_G(v)}\right)} \\
&\leq \frac{2 - \lambda_{n-k}}{C \cdot \log^3 n} \sum_{\substack{e=\{u,v\} \\ u \in A_i, v \in B_i}} w(u, v) \cdot \left(\frac{d_G(u) + d_G(v)}{2}\right),
\end{aligned} \tag{B.4}$$

where the last step follows by the means inequality. Let $\{(A_i, B_i)\}_{i=1}^k$ be the optimal cluster where $\bar{\rho}(k)$ is attained for graph $G$. Recall that for every $k \in \mathbb{N}$, the $k$-way dual Cheeger constant is defined by

$$\bar{\rho}_G(k) = \max_{(A_1, B_1), \cdots, (A_k, B_k)} \min_{1 \leq i \leq k} \overline{\phi}_G(A_i, B_i).$$

Then, we have for every $1 \leq i \leq k$ that

$$\bar{\rho}_G(k) \leq \overline{\phi}_G(A_i, B_i) = \frac{2 w_G(A_i, B_i)}{\mathrm{vol}_G(A_i \cup B_i)},$$

which implies

$$\frac{\bar{\rho}_G(k)}{2} \cdot \mathrm{vol}_G(A_i \cup B_i) \leq \sum_{\substack{e=\{u,v\} \\ u \in A_i, v \in B_i}} w_G(u, v). \tag{B.5}$$

Next, by the Chebyshev's inequality we have for any constant $c \in \mathbb{R}^+$ that

$$\mathbf{P}\left[|X - \mathbf{E}[X]| \geq c \cdot \mathbf{E}[X]\right]$$

$$\leq \frac{\mathbf{E}[X^2]}{c^2 \cdot \mathbf{E}[X]^2}$$

$$\leq \frac{\frac{2-\lambda_{n-k}}{C \cdot \log^3 n}\left(\sum_{\substack{e=\{u,v\} \\ u \in A_i, v \in B_i}} w_G(u,v) \cdot \left(\frac{d_G(u)+d_G(v)}{2}\right)\right)}{0.01 \cdot \left(\sum_{\substack{e=\{u,v\} \\ u \in A_i, v \in B_i}} w_G(u,v)\right)^2}$$

$$\leq \frac{\frac{2-\lambda_{n-k}}{C \cdot \log^3 n}\left(\sum_{\substack{e=\{u,v\} \\ u \in A_i, v \in B_i}} w_G(u,v) \cdot \left(\frac{d_G(u)+d_G(v)}{2}\right)\right)}{c^2 \cdot \left(\frac{\bar{\rho}_G(k)}{2} \cdot \mathrm{vol}_G(A_i \cup B_i)\right)^2} \tag{B.6}$$

$$= \frac{2 \cdot (2-\lambda_{n-k})}{c^2 \cdot C \cdot \log^3 n \cdot \bar{\rho}_G(k)^2} \cdot \frac{\sum_{\substack{e=\{u,v\} \\ u \in A_i, v \in B_i}} w_G(u,v) \cdot (d_G(u)+d_G(v))}{\mathrm{vol}_G(A_i \cup B_i)^2}$$

$$\leq \frac{2 \cdot (2-\lambda_{n-k})}{c^2 \cdot C \cdot \log^3 n \cdot \bar{\rho}_G(k)^2} \cdot \frac{\left(\max_{\substack{e=\{u,v\} \\ u \in A_i, v \in B_i}} \{d_G(u)+d_G(v)\}\right) \cdot \sum_{\substack{e=\{u,v\} \\ u \in A_i, v \in B_i}} w_G(u,v)}{\mathrm{vol}_G(A_i \cup B_i)^2}.$$

Since $\mathrm{vol}_G(A_i \cup B_i) = \sum_{u \in A_i} d_G(u) + \sum_{v \in B_i} d_G(v)$ and $d_G(u) = \sum_{u \sim v} w_G(u,v)$, we have

$$\max_{\substack{e=\{u,v\} \\ u \in A_i, v \in B_i}} \{d_G(u)+d_G(v)\} \leq \sum_{u \in A_i} d_G(u) + \sum_{v \in B_i} d_G(v) = \mathrm{vol}_G(A_i \cup B_i)$$

and

$$\sum_{\substack{e=\{u,v\} \\ u \in A_i, v \in B_i}} w_G(u,v) \leq \mathrm{vol}_G(A_i \cup B_i).$$

Thus, we have by (B.6) and the assumption of $\bar{\rho}(k) \geq \frac{1}{\log(n)}$ that

$$\mathbf{P}\left[|X - \mathbf{E}[X]| \geq c \cdot \mathbf{E}[X]\right] \leq \frac{2(2-\lambda_{n-k})}{c^2 \cdot C \cdot \log^3 n \cdot \bar{\rho}(k)^2} = O\left(\frac{1}{\log n}\right).$$

Hence, by choosing a sufficient large constant $c$ and the union bound, we have that

$$w_H(A_i, B_i) = \Omega\left(w_G(A_i, B_i)\right) \text{ for all } 1 \leq i \leq k. \tag{B.7}$$

Next, we show that the degree of every vertex in $H$ is approximately preserved with high probability. Based on the random variable $Y_e$ defined in (B.2), we define the random variable $Z_u$ by

$$Z_u = \sum_{v:v \sim u} Y_e.$$

Then, the expected value of $Z_u$ is given by

$$\mathbf{E}[Z_u] = \sum_{v:v \sim u} \mathbf{E}[Y_e] = \sum_{v:v \sim u} p_e \cdot \frac{w_G(u,v)}{p_e} = \sum_{v:v \sim u} w_G(u,v) = d_G(u),$$

and the second moment can be upper bounded by

$$\sum_{v:v \sim u} \mathbf{E}\left[Y_e^2\right] = \sum_{v:v \sim u} p_e \cdot \left(\frac{w_G(u,v)}{p_e}\right)^2 = \sum_{v:v \sim u} \frac{w_G(u,v)^2}{p_e} \leq \sum_{v:v \sim u} \frac{w_G(u,v)^2}{p_u(v)},$$

since $p_e \geq p_u(v)$. Now using the value of $p_u(v)$ from (3.1), we have

$$\sum_{v:v \sim u} \mathbf{E}\left[Y_e^2\right] \leq \sum_{v:v \sim u} w(u,v)^2 \cdot \frac{d_G(u) \cdot (2-\lambda_{n-k})}{w(u,v) \cdot C \cdot \log^3 n} = \frac{d_G(u) \cdot (2-\lambda_{n-k})}{C \cdot \log^3 n} \sum_{v:v \sim u} w_G(u,v)$$

$$= \frac{d_G^2(u) \cdot (2-\lambda_{n-k})}{C \cdot \log^3 n}$$

and for any edge $e = \{u, v\}$ we have that

$$0 \leq \frac{w(u,v)}{p_e} \leq \frac{w(u,v)}{p_u(v)} \leq \frac{d_G(u) \cdot (2 - \lambda_{n-k})}{C \cdot \log^3 n}.$$

Now, applying Bernstein's inequality (Lemma 11), we have

$$\mathbf{P}\left[|d_H(u) - d_G(u)| \geq \frac{d_u}{2}\right] = \mathbf{P}\left[|Z_u - E[Z_u]| \geq \frac{\mathbf{E}[Z_u]}{2}\right]$$

$$\leq 2 \cdot \exp\left(\frac{-\frac{1}{8} \cdot d_G^2(u)}{\frac{d_G^2(u) \cdot (2-\lambda_{n-k})}{C \cdot \log^3 n} + \frac{1}{6} \cdot \frac{d_G^2(u) \cdot (2-\lambda_{n-k})}{C \cdot \log^3 n}}\right)$$

$$= 2 \cdot \exp\left(-\frac{\frac{1}{8} \cdot C \cdot \log^3 n}{\frac{7}{6} \cdot (2 - \lambda_{n-k})}\right)$$

$$= o\left(\frac{1}{n^2}\right).$$

Hence, it holds by the union bound that, with high probability, the degree of all the vertices in $H$ are approximately preserved up to a constant factor. This implies that for any subset $S \subseteq V$, we have

$$\mathrm{vol}_H(S) = \Theta\left(\mathrm{vol}_G(S)\right),$$

more specifically,

$$\mathrm{vol}_H(A_i \cup B_i) = \Theta\left(\mathrm{vol}_G(A_i \cup B_i)\right), \tag{B.8}$$

for all $1 \leq i \leq k$. Thus, combining (B.7) and (B.8) gives us that

$$\overline{\phi}_H(A_i, B_i) = \Omega\left(\overline{\phi}_G(A_i, B_i)\right) \tag{B.9}$$

for all $1 \leq i \leq k$, which implies that

$$\bar{\rho}_H(k) \geq \min_{1 \leq i \leq k} \overline{\phi}_H(A_i, B_i) = \min_{1 \leq i \leq k} \Omega\left(\overline{\phi}_G(A_i, B_i)\right) = \Omega\left(\bar{\rho}_G(k)\right),$$

where the last equality follows from the fact that $\{(A_i, B_i)\}_{i=1}^k$ is the optimal cluster where $\bar{\rho}(k)$ is attained for graph $G$.

Next, we show that the top $(n-k)$-eigenspaces of $\mathcal{J}_G$ are preserved in $H$. Without loss of generality we assume the graph is connected. Since $\mathcal{J}_G = 2I - \mathcal{L}_G$ by definition, it holds that

$$\lambda_i(\mathcal{J}_G) = 2 - \lambda_{n+1-i}(\mathcal{L}_G). \tag{B.10}$$

Let

$$\mathcal{P} \triangleq \sum_{i=1}^{n-k}(2 - \lambda_i(\mathcal{L}_G))f_i f_i^{\mathsf{T}},$$

and with slight abuse of notation we call $\mathcal{P}^{-1/2}$ as the square root of the pseudo-inverse of $\mathcal{P}$, i.e.,

$$\mathcal{P}^{-1/2} = \sum_{i=1}^{n-k}(2 - \lambda_i(\mathcal{L}_G))^{-1/2} f_i f_i^{\mathsf{T}}.$$

Let $\overline{\mathcal{P}}$ be the projection on the spam of $\{f_1, f_2, \cdots, f_{n-k}\}$, then

$$\overline{\mathcal{P}} = \sum_{i=1}^{n-k} f_i f_i^{\mathsf{T}}.$$

Recall that, for each vertex $v$, the indicator vector $\chi_v \in \mathbb{R}^n$ is defined by $\chi_v(u) = \frac{1}{\sqrt{d_G(v)}}$ if $u = v$ and $\chi_v(u) = 0$ otherwise. For each edge $e = \{u, v\}$ of $G$ we define a vector $g_e = \chi_u + \chi_v \in \mathbb{R}^n$ and a random matrix $X_e \in \mathbb{R}^{n \times n}$ by

$$X_e = \begin{cases} w_H(u,v) \cdot \mathcal{P}^{-1/2} g_e g_e^{\mathsf{T}} \mathcal{P}^{-1/2} & \text{if } e = \{u, v\} \text{ is sampled by the algorithm,} \\ \mathbf{0} & \text{otherwise.} \end{cases} \tag{B.11}$$

Then, it holds that

$$\sum_{e \in E} X_e = \sum_{e=\{u,v\} \in F} w_H(u,v) \cdot \mathcal{P}^{-1/2} g_e g_e^\mathsf{T} \mathcal{P}^{-1/2}$$

$$= \mathcal{P}^{-1/2} \left( \sum_{e=\{u,v\} \in F} w_H(u,v) \cdot g_e g_e^\mathsf{T} \right) \mathcal{P}^{-1/2}$$

$$= \mathcal{P}^{-1/2} \mathcal{J}_H' \mathcal{P}^{-1/2},$$

where

$$\mathcal{J}_H' \triangleq \sum_{e=\{u,v\} \in F} w_H(u,v) \cdot g_e g_e^\mathsf{T}$$

is the signless Laplacian matrix of $H$ normalised with respect to the degree of the vertices in the original graph $G$. We will now prove that, with high probability the top $n - k$ eigenspaces of $\mathcal{J}_H'$ and $\mathcal{J}_G$ are approximately the same. We first analyse the expectation of $\sum_{e \in E} X_e$, and have that

$$\mathbf{E}\left[ \sum_{e \in E} X_e \right] = \sum_{e=\{u,v\} \in E} p_e \cdot w_H(u,v) \cdot \mathcal{P}^{-1/2} g_e g_e^\mathsf{T} \mathcal{P}^{-1/2}$$

$$= \sum_{e=\{u,v\} \in E} p_e \cdot \frac{w_G(u,v)}{p_e} \cdot \mathcal{P}^{-1/2} g_e g_e^\mathsf{T} \mathcal{P}^{-1/2}$$

$$= \mathcal{P}^{-1/2} \left( \sum_{e=\{u,v\} \in F} w_G(u,v) \cdot g_e g_e^\mathsf{T} \right) \mathcal{P}^{-1/2}$$

$$= \mathcal{P}^{-1/2} \mathcal{J}_G \mathcal{P}^{-1/2} = \sum_{i=1}^{n-k} f_i f_i^\mathsf{T} = \overline{\mathcal{P}}.$$

Moreover, for any edge $e = \{u, v\} \in E$ sampled by the algorithm, we have

$$\|X_e\| \le w_H(u,v) \cdot g_e^\mathsf{T} \mathcal{P}^{-1/2} \mathcal{P}^{-1/2} g_e = \frac{w_G(u,v)}{p_e} \cdot g_e^\mathsf{T} \mathcal{P}^{-1} g_e$$

$$\le \frac{w_G(u,v)}{p_e} \cdot \frac{1}{2 - \lambda_{n-k}} \cdot \|g_e\|^2$$

$$\le \frac{2 w_G(u,v)}{p_u(v) + p_v(u)} \cdot \frac{1}{2 - \lambda_{n-k}} \cdot \left( \frac{1}{d_G(u)} + \frac{1}{d_G(v)} \right)$$

$$\le \frac{2}{C \cdot \log^3 n},$$

where the second inequality follows by the min-max theorem of eigenvalues. Now we apply the matrix Chernoff bound (Lemma 12) to analyze the eigenvalues of $\sum_{e \in E} X_e$. Following Lemma 12 we set the parameters as follows:

$$\mu_{\max} = \lambda_{\max}\left( \mathbf{E}\left[ \sum_{e \in E} X_e \right] \right) = \lambda_{\max}\left( \overline{\mathcal{P}} \right) = 1,$$

$$R = \frac{2}{C \cdot \log^3 n}, \text{ and} \tag{B.12}$$

$$\delta = \frac{1}{2}.$$

Then using the Matrix Chernoff bound (Lemma 12), we have

$$\mathbf{P}\left[ \lambda_{\max}\left( \sum_{e \in E} X_e \right) \ge \frac{3}{2} \right] \le n \cdot \left( \frac{e^{\frac{1}{2}}}{1.5^{\frac{3}{2}}} \right)^{\frac{C \cdot \log^3 n}{2}} = O\left( \frac{1}{n^3} \right),$$

for some constant $C$; this implies that

$$\mathbf{P}\left[\lambda_{\max}\left(\sum_{e \in E} X_e\right) \leq \frac{3}{2}\right] = 1 - O\left(\frac{1}{n^3}\right). \tag{B.13}$$

On the other hand, since $\mathbf{E}\left[\sum_{e \in E} X_e\right] = \overline{\mathcal{P}}$, we have $\mu_{\min} = 1$ and hence keeping $R$ and $\delta$ the same as above, using the Matrix Chernoff bound (Lemma 12), we get

$$\mathbf{P}\left[\lambda_{\min}\left(\sum_{e \in E} X_e\right) \leq \frac{1}{2}\right] \leq n \cdot \left(\frac{e^{-\frac{1}{2}}}{0.5^{\frac{1}{2}}}\right)^{\frac{C \cdot \log^3 n}{2}} = O\left(\frac{1}{n^3}\right);$$

this implies that

$$\mathbf{P}\left[\lambda_{\min}\left(\sum_{e \in E} X_e\right) \geq \frac{1}{2}\right] = 1 - O\left(\frac{1}{n^3}\right). \tag{B.14}$$

Combining (B.13), (B.14) and the fact that $\sum_{e \in E} X_e = \mathcal{P}^{-1/2} \mathcal{J}'_H \mathcal{P}^{-1/2}$, with probability $1 - O\left(\frac{1}{n^3}\right)$ it holds for any non-zero $x \in \mathbb{R}^n$ in span$\{f_1, f_2, \cdots, f_{n-k}\}$ that

$$\frac{x^\mathsf{T} \mathcal{P}^{-1/2} \mathcal{J}'_H \mathcal{P}^{-1/2} x}{x^\mathsf{T} x} \in \left[\frac{1}{2}, \frac{3}{2}\right]. \tag{B.15}$$

Let $y = \mathcal{P}^{-1/2} x$, and we rewrite (B.15) as

$$\frac{y^\mathsf{T} \mathcal{J}'_H y}{y^\mathsf{T} \mathcal{P} y} = \frac{y^\mathsf{T} \mathcal{J}'_H y}{y^\mathsf{T} y} \cdot \frac{y^\mathsf{T} y}{y^\mathsf{T} \mathcal{P} y} \in \left[\frac{1}{2}, \frac{3}{2}\right].$$

Since $\dim(\text{span}\{f_1, f_2, \cdots, f_{n-k}\}) = n - k$, there exist $n - k$ orthogonal vectors whose Rayleigh quotient with respect to $\mathcal{J}'_H$ is $\Theta(\lambda_{n-k}(2I - \mathcal{L}_G))$. Hence, by the Courant-Fischer Theorem (Theorem 10) we have

$$\frac{1}{2} \cdot \lambda_{n-k}(2I - \mathcal{L}_G) \leq \lambda_{k+1}(\mathcal{J}'_H) \leq \frac{3}{2} \cdot \lambda_{n-k}(2I - \mathcal{L}_G) \tag{B.16}$$

By the definition of $\mathcal{J}'_H = D_G^{-1/2}\left(D_H + A_H\right) D_G^{-1/2}$, we have

$$\mathcal{J}_H = D_H^{-1/2}\left(D_H + A_H\right) D_H^{-1/2} = D_H^{-1/2}\left(D_G^{1/2} \cdot \mathcal{J}'_H \cdot D_G^{1/2}\right) D_H^{-1/2}.$$

Hence, we set $y = D_G^{1/2} D_H^{-1/2} x$ for any $x \in \mathbb{R}^n$ and have that

$$\frac{x^\mathsf{T} \mathcal{J}_H x}{x^\mathsf{T} \cdot x} = \frac{x^\mathsf{T} \cdot D_H^{-1/2}\left(D_G^{1/2} \cdot \mathcal{J}'_H \cdot D_G^{1/2}\right) D_H^{-1/2} \cdot x}{x^\mathsf{T} \cdot x} = \frac{y^\mathsf{T} \cdot \mathcal{J}'_H \cdot y}{x^\mathsf{T} \cdot x} \geq \frac{1}{2} \cdot \frac{y^\mathsf{T} \cdot \mathcal{J}'_H \cdot y}{y^\mathsf{T} \cdot y}, \tag{B.17}$$

where we use the fact that the degree of a vertex differs by a constant factor between $H$ and $G$. Similarly, we also have

$$\frac{x^\mathsf{T} \cdot \mathcal{J}_H \cdot x}{x^\mathsf{T} \cdot x} \leq \frac{3}{2} \cdot \frac{y^\mathsf{T} \cdot \mathcal{J}'_H \cdot y}{y^\mathsf{T} \cdot y}, \tag{B.18}$$

Let $T \subset \mathbb{R}^n$ be a $(k+1)$-dimensional subspace of $\mathbb{R}^n$ satisfying

$$\lambda_{k+1}(\mathcal{J}_H) = \max_{x \neq 0, x \in T} \frac{x^\mathsf{T} \cdot \mathcal{J}_H \cdot x}{x^\mathsf{T} \cdot x},$$

and $\widetilde{T} = \left\{D_G^{1/2} D_H^{-1/2} x : x \in T\right\}$. Since $D_G^{1/2} D_H^{-1/2}$ has full rank, $\widetilde{T}$ is also a $(k+1)$-dimensional subspace of $\mathbb{R}^n$. Hence, by the Courant-Fischer Theorem (Theorem 10) and (B.17), we have that

$$\lambda_{k+1}(\mathcal{J}'_H) = \min_{\substack{S \\ \dim(S)=k+1}} \max_{y \in S \setminus \{\mathbf{0}\}} \frac{y^\mathsf{T} \cdot \mathcal{J}'_H \cdot y}{y^\mathsf{T} \cdot y}$$

$$\leq \max_{y \in \widetilde{T} \setminus \{\mathbf{0}\}} \frac{y^\mathsf{T} \cdot \mathcal{J}'_H \cdot y}{y^\mathsf{T} \cdot y} \tag{B.19}$$

$$\leq 2 \cdot \max_{x \in T \setminus \{\mathbf{0}\}} \frac{x^\mathsf{T} \cdot \mathcal{J}_H \cdot x}{x^\mathsf{T} \cdot x} = 2 \cdot \lambda_{k+1}(\mathcal{J}_H).$$

Next, using (B.16) and (B.19), we have

$$\frac{1}{2} \cdot \lambda_{k+1}(\mathcal{J}_G) \leq \lambda_{k+1}(\mathcal{J}_H') \leq 2 \cdot \lambda_{k+1}(\mathcal{J}_H),$$

which implies that

$$\frac{1}{4} \cdot \lambda_{k+1}(\mathcal{J}_G) \leq \lambda_{k+1}(\mathcal{J}_H). \tag{B.20}$$

Similarly, let $U \subset \mathbb{R}^n$ be an $(n-k)$-dimensional subspace of $\mathbb{R}^n$ satisfying

$$\lambda_{k+1}(\mathcal{J}_H) = \min_{x \neq 0, x \in U} \frac{x^\mathsf{T} \cdot \mathcal{J}_H \cdot x}{x^\mathsf{T} \cdot x},$$

and $\widetilde{U} = \left\{ D_G^{1/2} D_H^{-1/2} x : x \in U \right\}$. Since $D_G^{1/2} \cdot D_H^{-1/2}$ has full rank, $\widetilde{U}$ is also an $(n-k)$-dimensional subspace of $\mathbb{R}^n$. Thus, using the Courant-Fischer Theorem (Theorem 10) and (B.18), we have

$$\begin{aligned}
\lambda_{k+1}(\mathcal{J}_H') &= \max_{\substack{S \\ \dim(S)=n-k}} \min_{y \in S \setminus \{\mathbf{0}\}} \frac{y^\mathsf{T} \cdot \mathcal{J}_H' \cdot y}{y^\mathsf{T} \cdot y} \\
&\geq \min_{y \in \widetilde{U} \setminus \{\mathbf{0}\}} \frac{y^\mathsf{T} \cdot \mathcal{J}_H' \cdot y}{y^\mathsf{T} \cdot y} \\
&\geq \frac{2}{3} \cdot \min_{x \in U \setminus \{\mathbf{0}\}} \frac{x^\mathsf{T} \cdot (2I - \mathcal{L}_H) \cdot x}{x^\mathsf{T} \cdot x} \\
&= \frac{2}{3} \cdot \lambda_{k+1}(\mathcal{J}_H).
\end{aligned} \tag{B.21}$$

Next, by (B.16) and (B.21) we have

$$\frac{2}{3} \cdot \lambda_{k+1}(\mathcal{J}_H) \leq \gamma_{k+1}(\mathcal{L}_H') \leq \frac{3}{2} \cdot \lambda_{k+1}(\mathcal{J}_G),$$

which implies that

$$\lambda_{k+1}(\mathcal{J}_H) \leq \frac{9}{4} \cdot \lambda_{k+1}(\mathcal{J}_G). \tag{B.22}$$

Thus, combining (B.20) and (B.22) we have

$$\frac{1}{4} \cdot \lambda_{k+1}(\mathcal{J}_G) \leq \lambda_{k+1}(\mathcal{J}_H) \leq \frac{9}{4} \cdot \lambda_{k+1}(\mathcal{J}_G),$$

Hence, the the top $n - k$ eigenspaces of $\mathcal{J}_G$ are preserved in $\mathcal{J}_H$. This proves the second statement of the theorem.

## C    OMITTED DETAIL FROM SECTION 4

In this section we list all the proofs omitted from Section 4.

*Proof of Lemma 6.* The proof follows from Macgregor & Sun (2021), which proves the result for undirected graphs. We include the proof here for completeness. Let $S = A_1 \cup B_2$ in $H$, then

$$\begin{aligned}
\phi_H(A_1 \cup B_2) = \phi_H(S) &= \frac{w_H(S, V \setminus S)}{\mathrm{vol}_H(S)} \\
&= \frac{\mathrm{vol}_H(S) - 2w_H(S, S)}{\mathrm{vol}_H(S)} \\
&= 1 - \frac{2w_H(S, S)}{\mathrm{vol}_H(S)} = 1 - \frac{2w_{\overrightarrow{G}}(A, B)}{\mathrm{vol}_{\mathrm{out}}(A) + \mathrm{vol}_{\mathrm{in}}(B)} = f_{\overrightarrow{G}}(A, B).
\end{aligned} \tag{C.1}$$

This proves the first statement of the lemma. The second statement of the lemma follows by the similar argument. $\qquad \square$

*Proof of Lemma 7.* By definition, we have that

$$f_{\overrightarrow{G}}(A, B) = 1 - \overline{\phi}_{\overrightarrow{G}}(A, B), \tag{C.2}$$

and this implies that

$$
\begin{aligned}
\bar{\rho}_{\overrightarrow{G}}(k) &= \max_{(A_1,B_1),\ldots,(A_k,B_k)} \min_{1 \leq i \leq k} \overline{\phi}_{\overrightarrow{G}}(A_i, B_i) \\
&= \max_{(A_1,B_1),\ldots,(A_k,B_k)} \min_{1 \leq i \leq k} \left(1 - f_{\overrightarrow{G}}(A_i, B_i)\right) \\
&= 1 - \min_{(A_1,B_1),\ldots,(A_k,B_k)} \max_{1 \leq i \leq k} f_{\overrightarrow{G}}(A_i, B_i) \\
&= 1 - \min_{C_1,\ldots,C_k} \max_{1 \leq i \leq k} \phi_H(C_i),
\end{aligned}
$$

where the second line follow by (C.2), and the last one follows by Lemma 6 and $C_i = A_{i_1} \cup B_{i_2}$. $\quad\square$

