# OpenReview forum: "Efficient Sparsification of Densely Connected Clusters"
_ICLR.cc/2025/Conference — Submitted to ICLR 2025_

### Official Review · Reviewer_Q1WE · 2024-10-25

**Soundness:** 3
**Presentation:** 2
**Contribution:** 2
**Rating:** 5
**Confidence:** 2

**Summary:**

This paper proposes a sparsification method that maintain the strucutre of clusters for undirected and directed graphs. Specifically, the authors present two nearly-linear algorithms. Also, the authors provide a thorough theoretical analysis and expereimental results using synthetic datasets.

**Strengths:**

- Thorough theoretical analysis.

**Weaknesses:**

- What are the practical advantages of addressing the considered problem? What is the motivation?
- Some notations such as $\omega_G$ and $vol_G$ are used without being clearly defined.
- The algorithms assume that the input graphs exhibit clearly defined cluster structures. What if the clusters are less well-defined? Does it impact the effectiveness of sparsification?
- The experiments are conducted using only synthetic datasets. The effectiveness of real-world datasets are not tested.
- How sensitive are the algorithms with respect to the parameters?

**Questions:**

Please find the questions in the **weaknesses**.

---

> ### Author Response · Authors · 2024-11-21
>
> Thank you for the time reviewing our paper. Here is our response to your questions:
>
> > **Weakness 1:** What are the practical advantages of addressing the considered problem? What is the motivation?
>
> **Response:** While traditional clustering objective is to find clusters of low conductance, many recent studies examine the structure of clusters, in particular two vertex sets that are  densely connected between each other and relatively loosely connected to the rest of the graph. For example, when employing graphs to model a migration dataset, we'd like to find two sets of counties (two vertex sets) between which people move more frequently. Similar applications appear often in analysing many real-world datasets, and we need to focus on how clusters are connected between each other rather than simply finding vertex sets of low conductance.  The importance of this line of research can be also witnessed by the work of Macgergor and Sun (ICML'21), which is selected as a long talk of the conference, and the reference within.
>
> > **Weakness 2:** Some notations such as $\omega_G$  and $vol_G$ are used without being clearly defined.
>
> **Response:** Thank you for pointing this out. We will clearly define all the used notations in the next version of our paper.
>
> > **Weakness 3:**  The algorithms assume that the input graphs exhibit clearly defined cluster structures. What if the clusters are less well-defined? Does it impact the effectiveness of sparsification?
>
> **Response:** As shown in Theorem 1 and 2 of the submission, the quality of our output clusters is quantified with respect to the optimal clusters of the underlying graph, and their bipartiteness ratios are within a constant factor to each other. Hence, even if the clusters are less well-defined in the input graph, we guarantee that the output of our algorithm constant-factor approximates the optimal ones with respect to the bipartiteness ratio.
>
> > **Weakness 4:** The experiments are conducted using only synthetic datasets. The effectiveness of real-world datasets are not tested.
>
>
> **Response:** In the common response to all the reviewers, we have added the experimental result on US migration  dataset.
>
> > **Weakness 5:** How sensitive are the algorithms with respect to the parameters?
>
>
> **Response:** All the parameters of our algorithm are only used to determine the sampling probability. Reasonable estimation of these parameters suffices for our purpose, and this will mainly influence the number of sampled edges. From this perspective, our presented algorithms are robust with respect to the parameters.

---

> > ### Comment · Reviewer_Q1WE · 2024-11-26
> >
> > I appreciate the responses from the authors, and they have addressed some of my concerns. However, regarding W4, while I appreciate the authors for testing on a real-world dataset, I belive testing on only one dataset is insufficient to demonstrate the generality and applicability of the method. Also, regarding W5, I think parameter sensitivity analysis requires more rigorous empirical validation. Thus, I would maintain my score.

---

### Official Review · Reviewer_5rub · 2024-11-01

**Soundness:** 2
**Presentation:** 1
**Contribution:** 2
**Rating:** 5
**Confidence:** 3

**Summary:**

The sparsification algorithm presented in this paper effectively handles densely connected clusters in both undirected and directed graphs. The article provides sufficient proofs demonstrating the method's time efficiency. However, the writing is somewhat confusing, making it difficult to clearly explain the approach used. Additionally, the experiments lack relevant data validation and comparative analysis.

**Strengths:**

1.The paper offers sufficient theoretical foundation with detailed analysis of Cheeger constants and spectral theory, providing solid proof of the algorithm's effectiveness.

2.The sparsification algorithm significantly speeds up existing clustering methods while effectively handling densely connected clusters in both undirected and directed graphs.

3.The algorithm is applicable to both undirected and directed graphs, demonstrating its flexibility and usefulness across different types of graph structures.

**Weaknesses:**

1. The structure of the article is confusing, and the method is not clearly explained. The article should highlight its contributions and clearly describe the solution ideas and working methods. Additionally, the use of symbols and lemmas, as well as the arrangement of proofs, should be more detailed.
    - For example, what’s vol_{G} and w_{G} is not explained in the intro when Equations 1.1 is introduced.
    - A_i \cup B_i \neq \emptyset for different i,j \in [k], typo?
2. The experiments lack relevant work. The absence of extensive testing with different algorithms on multiple datasets may impact the algorithm's universality and reliability.
3. The experiments are only conducted on synthetic datasets without real-world datasets. It would be better to provide more experiments on real-world datasets.
4. While the algorithm shows improved efficiency empirically, the trends may need further explanation. For example, the improvement seems larger when the dataset is larger, while on directed graphs the improvement seems smaller when the dataset is larger (with 2000 and 5000 vertices, respectively). Further explanations are needed here, whether there are some gaps between the implementation and theoretical analysis.

**Questions:**

How do you plan to improve the structure of the article to enhance clarity? Will you include more detailed descriptions of the solution ideas and working methods?

Can you explain why the 5000-vertex setting in Figure 3 (a) takes less time compared to 4000-vertex setting? Will you consider testing the effectiveness of this method on a real-world dense dataset?

---

> ### Author Response · Authors · 2024-11-21
>
> Thank you for the careful reading and evaluation of our submission.
>
> > **Question 1:** How do you plan to improve the structure of the article to enhance clarity? Will you include more detailed descriptions of the solution ideas and working methods?
>
> **Response:** We will improve the presentation of the paper, making sure that all the notations are formally defined before they're used the first time. We will further include more discussions of the importance of the problem, and the main ideas behind the design of our algorithms.
>
> > **Question 2:** Can you explain why the 5000-vertex setting in Figure 3 (a) takes less time compared to 4000-vertex setting? Will you consider testing the effectiveness of this method on a real-world dense dataset?
>
> **Response:** This is an outstanding question. Notice that our input graphs are constructed such that every cluster is a random graph, and the two clusters are connected by random edges; moreover, any pair of vertices from different clusters is connected with higher probability. Hence, sampling a good fraction of the input graph might result in two clusters with roughly the same bipartiteness ratio as the optimal one. Moreover, following our sparsification procedure we apply the MS algorithm, which is a local algorithm and doesn't necessarily explore the entire graph; the MS algorithm terminates once it finds the two clusters with a certain guarantee. This is why the 5,000-vertex setting takes roughly the same time compared with the 4,000-vertex setting.

---

> > ### Comment · Reviewer_5rub · 2024-11-25
> >
> > Thanks for the response.

---

### Official Review · Reviewer_LpPM · 2024-11-01

**Soundness:** 2
**Presentation:** 1
**Contribution:** 1
**Rating:** 3
**Confidence:** 3

**Summary:**

The paper proposes efficient sparsification algorithms to preserve densely connected vertex sets in undirected and directed graphs.

**Strengths:**

The paper proposes efficient sparsification algorithms to preserve densely connected vertex sets in both undirected and directed graphs.

**Weaknesses:**

1. Notations (e.g., w_G(V_1, V_2), vol_G(V_1 \union V_2)) and concepts (e.g., k pairs of densely connected clusters) are introduced in the introduction but are not defined.

2. The significance of the proposed algorithms is not clear.

3. The experimental results are not convincing.

**Questions:**

Notations (e.g., w_G(V_1, V_2), vol_G(V_1 \union V_2)) and concepts (e.g., k pairs of densely connected clusters) are introduced in the introduction but are not defined, and therefore I was confused right from the beginning when reading the paper. In fact, I still do not fully understand what "k pairs of densely connected clusters" actually means, I can understand there are k disjoint subsets of vertices in the graph and the induced subgraph of each subset of vertex may form a cluster (i.e., a densely connected subgraph), but where do you get k pairs of densely connected clusters from k disjoint subsets of vertices? Therefore, it is very difficult for me to understand the paper when the important concepts are not properly defined. I suggest the authors to clearly define all notations and concepts the first time they are introduced in the paper.

The significance of the proposed algorithms is not clear, i.e., why the results are important theoretically and practically?  It would be good if the authors could clearly explain the significance of the proposed method using a running example, e.g., given an input graph, what does the algorith output and show that the output is important.

The experiments were conducted on synthetic graphs only with k = 2. It would be more convincing to show the results on representative real graphs and large values of k. Another question is that k is actually unknown in practice, how do the authors address this problem?

---

> ### Author Response · Authors · 2024-11-21
>
> Thank you for the time evaluating our submission. Here is our response to your questions:
>
> > **Question 1:** Notations (e.g., w_G(V_1, V_2), vol_G(V_1 \union V_2)) and concepts (e.g., k pairs of densely connected clusters) are introduced in the introduction but are not defined, and therefore I was confused right from the beginning when reading the paper. In fact, I still do not fully understand what "k pairs of densely connected clusters" actually means, I can understand there are k disjoint subsets of vertices in the graph and the induced subgraph of each subset of vertex may form a cluster (i.e., a densely connected subgraph), but where do you get k pairs of densely connected clusters from k disjoint subsets of vertices? Therefore, it is very difficult for me to understand the paper when the important concepts are not properly defined. I suggest the authors to clearly define all notations and concepts the first time they are introduced in the paper.
>
> **Response:** We will improve the presentation of the paper, and clearly define the notations before they are used the first time. We call a pair of vertex sets $A,B$ densely connected if most edges leaving $A$ and $B$ are the ones between $A$ and $B$, and there are few edges between $A\cup B$ and $V\setminus (A\cup B)$; i.e., $A,B$ form an almost bipartite component. We further say $G$ has $k$ pairs of densely connected clusters if $G$ contains $k$ such pairs of $(A,B)$. We will use one example to make this definition clearer in the next version of the paper.
>
> >**Question 2:** The significance of the proposed algorithms is not clear, i.e., why the results are important theoretically and practically? It would be good if the authors could clearly explain the significance of the proposed method using a running example, e.g., given an input graph, what does the algorithm output and show that the output is important.
>
> **Response:** We first discuss the significance of the problem. Notice that, while traditional clustering objective is to find clusters of low conductance, many recent studies examine the structure of clusters, in particular two vertex sets that are densely connected between each other and relatively loosely connected to the rest of the graph. For example, when employing graphs to model a migration dataset, we'd like to find two sets of counties (two vertex sets) between which people move more frequently. Similar applications appear often in analysing many real-world datasets, and we need to focus on how clusters are connected between each other rather than simply finding vertex sets of low conductance. The importance of this line of research can be also witnessed by the work of Macgergor and Sun (ICML'21), which is selected as a long talk of the conference, and the reference within. We view our result important since this is the first such result showing that one can effectively sparsity the input graph while preserving this type of cluster structures.
>
> To further provide a running example, in the response to all the reviewers we present an experimental result on the US migration dataset. Here, every county is represented as a vertex, and the weight between any pair of vertices is the number of people moving from one county to the other during a certain period of time. Then, the $k$ pairs of clusters correspond to the $k$ groups of regions (several conties), in which people tend to move form one region to another. This pair of densely connected clusters is usually viewed as the higher-order information of cluster structure, and has many other applications beyond our shown-cased migration data analysis.
>
> >**Question 3:** The experiments were conducted on synthetic graphs only with k = 2. It would be more convincing to show the results on representative real graphs and large values of k. Another question is that k is actually unknown in practice, how do the authors address this problem?
>
> **Response:**  On the lack of multiple clusters case for experiments, notice that we applied the existing local algorithms following our sparsification algorithms in order to measure the quality of our output; these local algorithms only explore part of the input graph, and find 2 densely connected clusters even the input graph has many pairs of densely connected clusters. However, this experiment setup guarantees that our experimental results can be easily generalised to the case of multiple clusters.
>
> We added one experimental result on the real-world dataset to support our claim.
>
> For the unknown value of $k$, notice that actually our algorithm doesn't need the value of $k$ as input, this can be viewed as an advantage of our algorithm.

---

### Official Review · Reviewer_58Yn · 2024-11-03

**Soundness:** 3
**Presentation:** 3
**Contribution:** 2
**Rating:** 5
**Confidence:** 4

**Summary:**

The authors propose a sparsification algorithm for dense clusters that approximately preserves the clustering the sparse representation of the underlying graph.

**Strengths:**

The authors study an interesting and well-studied problem.

**Weaknesses:**

1. A good fraction of the paper is rehashing well-known definitions.
2. This is a well-studied problem.  There are a lot of missing references including papers that compare different sparsification techniques (e.g., see https://arxiv.org/pdf/2311.12314 and references within).  Related references include :
        1) D.A. Spielman and N. Srivastava. Graph sparsification by effective resistances. SIAM Journal on Computing, 40(6), pp.1913-1926. 2011.
        2) http://cs-www.cs.yale.edu/homes/spielman/PAPERS/CACMsparse.pdf
        3) https://www.cs.ubc.ca/~nickhar/papers/Sparsifier/Sparsifier-Long.pdf. and references within.
        4) https://www.cis.upenn.edu/~sanjeev/papers/focs22_weighted_sparsification.pdf

**Questions:**

1. The papers I referred to above address the questions of the sparse graph satisfying some properties including community detection. Please differentiate how the proposed work differs from the earlier work.

Update:  The authors have sufficiently addressed my concerns.  I have raised my score. Even as the authors claim their approach works well for the directed case as well, they have not conclusively demonstrated the effectiveness of their approach practically. I am not able to strongly back the paper as the work still seems lacking sufficiently new ideas for a well-studied problem.

---

> ### Author Response · Authors · 2024-11-21
>
> > "The papers I referred to above address the questions of the sparse graph satisfying some properties including community detection. Please differentiate how the proposed work differs from the earlier work."
>
> Thank you for the time reviewing our submission. However, we believe that you have  missed the contribution of our work. We are very familiar with the reference on spectral sparsification including the papers mentioned in your report, and  there are three key differences  between our work and your mentioned reference:
>
> First of all, the objective of our problem is to preserve the cut values  $w(A, B)$ for **certain pairs of vertex sets $A$ and $B$**, while all the algorithms you mentioned can only preserve the cut value $w(S, V\setminus S)$ for any vertex set $S$.
>
> Secondly, our designed algorithms work for both undirected graphs and directed ones. But most graph sparsification algorithms only work for undirected graphs.
>
> Finally, while the design of most graph sparsification algorithms are based on Laplacian solvers making it unpractical, our designed algorithms only use  random sampling.
>
> Taking this into account, we sincerely hope that you can re-evaluate our submission. Thanks.

---

> > ### Comment · Reviewer_58Yn · 2024-11-27
> >
> > I thank the authors for clarifying their contributions.
> >
> > However, I don't completely follow your model of computing projected cuts (i.e., w.r.t. some subsets of vertices in the graph) that result in densely connected clusters.  Why is this is an interesting use case for finding densely connected clusters?  Secondly,  there are linear time solvers to for graph sparsification.  It will help if you can distinguish your work from those methods (in terms of efficiency and optimality loss).

---

> > > ### Author Response · Authors · 2024-11-27
> > >
> > > Thank you for reading our response. Here are our answers to your specific questions.
> > >
> > > > I don't completely follow your model of computing projected cuts (i.e., w.r.t. some subsets of vertices in the graph) that result in densely connected clusters. Why is this is an interesting use case for finding densely connected clusters?
> > >
> > > Computing the projected cuts that result in the densely connected clusters has been widely studied in both theoretical computer science and machine learning. For instance, in the theoretical side the projected cuts are used in designing approximation algorithms for the max cut problem (e.g., Trevisan, 2008); in the applied side, one specific example is to apply a graph to model a migration dataset: here we'd like to find two sets of counties (two vertex sets) between which people move more frequently. Similar applications appear often in analysing many real-world datasets, and we need to focus on how clusters are connected between each other rather than simply finding vertex sets of low conductance. The importance of this line of research can be also witnessed by the work of Macgergor and Sun (ICML'21), which is selected as a long talk of the conference, and the reference within.
> > >
> > > To further provide a running example, in the response to all the reviewers we present an experimental result on the US migration dataset. Here, every county is represented as a vertex, and the weight between any pair of vertices is the number of people moving from one county to the other during a certain period of time. Then, the  pairs of clusters correspond to the groups of regions (several conties), in which people tend to move form one region to another. This pair of densely connected clusters is usually viewed as the higher-order information of cluster structure, and has many other applications beyond our shown-cased migration data analysis.
> > >
> > > > Secondly, there are linear time solvers to for graph sparsification. It will help if you can distinguish your work from those methods (in terms of efficiency and optimality loss).
> > >
> > > First of all, all of the current linear-time solvers cannot be applied in our problem as they don't preserve projected cut values. For efficiency, our algorithm runs in nearly-linear time, which is the same as most state-of-the-art algorithms for constructing spectral sparsifiers. With respect to optimality, our constructed graphs have $\widetilde{O}(n)$ edges, which is the same as the guarantee achieved by Spielman-Teng, Spielman-Srivastava algorithms but slightly worse than the guarantee achieved by BSS and the subsequent improvement (Lee and Sun).
> > >
> > > Finally, our designed algorithms are not based on Laplacian solvers; hence, it is more practical than most spectral sparsification algorithms.

---

### Official Review · Reviewer_A7RN · 2024-11-04

**Soundness:** 4
**Presentation:** 3
**Contribution:** 3
**Rating:** 8
**Confidence:** 3

**Summary:**

The paper introduced sparsification algorithms and the corresponding theoretical analysis that preserve the structure of densely connected vertex sets in both undirected and directed scenario. The experiments on synthesis dataset indicates the efficiency of the proposed sparsification and the effectiveness for clustering.

**Strengths:**

1. The algorithm description is straightforward and easy to follow. For undirected graph, local sampling and reweighing the edges are the necessary steps for sparsification; for directed graph, the process to convert to undirected graph and converted back are the extra steps.
2. The theoretical analysis and the probability guarantee makes the methodology promising; the nearly-linear running time makes the algorithm applicable.
3. The specific challenges in directed graph case are discussed in detail.

**Weaknesses:**

1. Some of the notations could be more clear:
    i.  The w_G and vol_G in line 37 are not defined.
    ii. Although it mentions C is a universal constant on Lemma 3 and Lemma 4, it is still not clear how to determine the value of C in formula 3.1 for the experiment.
2. There is no experiment conducted on multiple cluster case or real-world dataset.

**Questions:**

1. Is there a way to adjust the level of sparsification? How will the efficiency and effectively be affected?

---

> ### Author Response · Authors · 2024-11-21
>
> Thank you for the time and positive evaluation on our submission. In the common response to all the reviewers, we have added additional experimental results on the US migration datasets. Here are our response to your questions:
>
> > **Weakness 2:** There is no experiment conducted on multiple cluster case or real-world dataset.
>
> **Response:** On the lack of multiple clusters case for experiments, notice that we applied the existing *local algorithms* following our sparsification algorithms in order to measure the quality of our output; these
> local algorithms only explore part of the input graph, and find
>  2 densely connected clusters even the input graph has many pairs of densely connected clusters. However, this experiment setup guarantees that our experimental results can be easily generalised to the case of multiple clusters.
>
> > **Question:** Is there a way to adjust the level of sparsification? How will the efficiency and effectively be affected?
>
> We're not very sure which type of level of sparsification you refer to, since our designed algorithms don't involve different levels. If you could explain your question a bit more, we're very happy to explore this direction and answer your question.

---

> > ### Comment · Reviewer_A7RN · 2024-11-25
> >
> > Thanks for the response.
> > I do not think the response answered my question.
> >
> > 1. The choose of C in formula 3.1 to calculate the probability for sample is not clear.
> > 2. The author claims "setup guarantees that our experimental results can be easily generalised to the case of multiple clusters.". In the paper, author also indicates the method works for "k pairs of densely-connected clusters". Then experiments on multiple clusters should be conducted. If no real-world dataset, the synthesis dataset could also help.
> > 3. The question is about the trade-off between between the level of sparsification and the algorithm's performance or accuracy. For example, in [1], Figure 2 discussed the relationship between level of sparsfication and error rate.
> >
> > [1] Bravo Hermsdorff, Gecia, and Lee Gunderson. "A unifying framework for spectrum-preserving graph sparsification and coarsening." Advances in Neural Information Processing Systems 32 (2019).

---

> > > ### Author Response · Authors · 2024-11-25
> > >
> > > Thank you for reading our response carefully. Here are our answers to your further questions:
> > >
> > > > *Question 1:* The choose of C in formula 3.1 to calculate the probability for sample is not clear.
> > >
> > > We didn't specify the constant here, since most constants would work. To see this, notice that this choice of $C$ only changes the sampling probability by a constant factor, and hence doesn't influence the asymptotic order of the sampled edges. On the other hand, the specific value of $C$ only changes the error probability when applying the Matrix Chernoff bound, which is a polynomially small in $n$.
> > >
> > > For our experiments we simply set $C=2$.
> > >
> > > > *Question 2:* The author claims "setup guarantees that our experimental results can be easily generalised to the case of multiple clusters.". In the paper, author also indicates the method works for "k pairs of densely-connected clusters". Then experiments on multiple clusters should be conducted. If no real-world dataset, the synthesis dataset could also help.
> > >
> > > Thank for for the follow-up question. To make our point clearer, let's assume that the input graph contains $k\geq 2$ pairs of densely-connected clusters, and with the designed algorithms we obtain a sparse graph consisting of $k$ pairs of densely-connected clusters. BUT, since a local algorithm that we apply after sparsification only explores part of the graph and returns a **single-pair** of clusters, the remaining part of the graph (apart from the region that a local algorithm explores) is unknown to the algorithm. Since we select a *random vertex* of the input graph to start the local algorithm and the local algorithm always returns a *single pair*, our experimental results hold even when the input graph has $k\geq 2$ clusters.
> > >
> > > > *Question 3:* The question is about the trade-off between between the level of sparsification and the algorithm's performance or accuracy. For example, in [1], Figure 2 discussed the relationship between level of sparsfication and error rate.
> > >
> > > Thank you for the additional reference and, if we understand it correctly, the level of sparsification refers to the sparsity of the resulting graph, i.e., the number of sampled edges, and this related to Question 1 you mentioned earlier.
> > >
> > > The constant $C$ that you pointed out in Question 1 does influence the total number of sampled edges, and the accuracy of the algorithm. Formally, with a larger value of $C$ the output graph of the algorithm has more edges and the output graph satisfies the approximation guarantee with higher probability. We will make it clearer in the next version of our submission.
> > >
> > >
> > > Please just let us know if more clarification on these points is needed.

---

### Author Response · Authors · 2024-11-21
**Experimental results on real-world datasets**

To respond to the reviewers' request, we conduct additional experiments on the real-word data sets. Specifically, we follow the work of Macgregor and Sun (2021), and evaluate the performance of our algorithm on the US migration dataset. Given this dataset, we construct the digraph as follows: every county in the mainland USA is represented by a vertex; for any vertices $i,j$, the edge weight of $(i,j)$ is given by $|(M_{i,j} − M_{j,i})/(M_{i,j}+M_{j,i})|$, where $M_{i,j}$ is the number of people who migrated from county $i$ to county $j$ between 1995 and 2000; in addition, the direction of $(i,j)$ is set to be from $i$ to $j$ if $M_{i,j} > M_{j,i}$, otherwise the direction is set to be the opposite.

We compare the output of ECD and the output of our sparsification algorithm + ECD. Furthermore, we use the vertices corresponding to different counties as the input of the local algorithm ECD. Our experimental result is as follows:

 |County Name               | Target $\phi$  |  ECD Runtime     |  ECD Flow-Ratio       |   ECD+Our  Runtime |   ECD+Our Flow-Ratio|
 | ----------- | ----------- |----------- |----------- |----------- |----------- |
 |Maricopa County  |0.1| 20.28   | 0.416      | 14.124  | 0.45    |
  |                 |0.2|20.661|0.414 |13.434 |0.417
  |Virginia Beach City| 0.1| 14.725  | 0.413      | 11.172  | 0.672|
|                    |0.2| 15.31 |0.546 | 12.29 | 0.621      |
|Kanawha county        | 0.1           | 10.083 | 0.33       | 8.459   | 0.364 |
|                  | 0.2| 9.318 | 0.33|8.483 | 0.33  |

These experimental results clearly demonstrate that with our sparsification algorithm the local algorithm achieves roughly the same flow ratio, and our sparsification clearly speeds up the total running time of the algorithm. Moreover, the runtime speedup is more significant when the local algorithm explores more parts of the input graph (corresponding to a higher value of $\phi$).

---

### Author Response · Authors · 2024-11-26

Dear reviewers, with the discussion period ending soon, we’d really value your feedback on our responses - let us know if there’s anything else we should clarify or fix. Thank you again for your time.

---

### Meta-Review · Area_Chair_e9nJ · 2024-12-17

**Metareview:**

This paper presents efficient sparsification algorithms for both undirected and directed graphs, designed to preserve the structure of densely connected clusters. The authors provide theoretical analysis demonstrating the efficiency of their algorithms, which have near-linear time complexity. They also present experimental results on synthetic datasets, showing the effectiveness of their approach for clustering tasks.

Reviewers appreciate the simplicity of the proposed algorithms and the clarity of the theoretical analysis. The near-linear time complexity is a desirable property for sparsification methods.

However, reviewers also identify some weaknesses:

- Limited Experimental Evaluation: The experiments are limited to synthetic datasets and do not include real-world data or scenarios with multiple clusters. This makes it difficult to assess the practical applicability of the proposed method.
- Missing Related Work: The paper lacks a comprehensive discussion of related work on graph sparsification and clustering. This limits the context for the contributions of the paper.

Recommendation:

While the paper presents efficient sparsification algorithms with theoretical guarantees, the reviewers agree that it does not meet the bar for acceptance at ICLR in its current form.

**Additional Comments On Reviewer Discussion:**

The discussion went on smoothly although even after it, the paper seems below the acceptance bar.

---

### Decision · Program_Chairs · 2025-01-22

Reject